# FROZEN TRANSFORMERS IN LANGUAGE MODELS ARE EFFECTIVE VISUAL ENCODER LAYERS

**Ziqi Pang**  **Ziyang Xie**\*  **Yunze Man**\*  **Yu-Xiong Wang**
University of Illinois Urbana-Champaign
{ziqip2,ziyang8,yunzem2,yxw}@illinois.edu
https://github.com/ziqipang/LM4VisualEncoding

## ABSTRACT

This paper reveals that large language models (LLMs), despite being trained solely on text data, are *surprisingly* strong encoders for *purely* visual tasks in the absence of language. Even more intriguingly, this can be achieved by a simple yet previously overlooked strategy – employing a *frozen* transformer block from *pre-trained* LLMs as a constituent encoder layer to directly process visual tokens. Our work pushes the boundaries of leveraging LLMs for computer vision tasks, significantly departing from conventional practices that typically necessitate a multi-modal vision-language setup with associated language prompts, inputs, or outputs. We demonstrate that our approach consistently enhances performance across *a diverse range of tasks*, encompassing purely 2D and 3D visual recognition tasks (*e.g.*, image and point cloud classification), temporal modeling tasks (*e.g.*, action recognition), non-semantic tasks (*e.g.*, motion forecasting), and multi-modal tasks (*e.g.*, 2D/3D visual question answering and image-text retrieval). Such improvements are a general phenomenon, applicable to various types of LLMs (*e.g.*, LLaMA and OPT) and different LLM transformer blocks.

We additionally propose the *information filtering* hypothesis to explain the effectiveness of pre-trained LLMs in visual encoding – the pre-trained LLM transformer blocks discern informative visual tokens and further amplify their effect. This hypothesis is empirically supported by the observation that the feature activation, after training with LLM transformer blocks, exhibits a stronger focus on relevant regions. We hope that our work inspires new perspectives on utilizing LLMs and deepening our understanding of their underlying mechanisms.

## 1 INTRODUCTION

Large language models (LLMs), trained on massive amounts of text data, have recently demonstrated remarkable potential across various tasks, extending beyond their original linguistic domain. For example, in the field of computer vision, LLMs exhibit the ability to interact with visual tokens and *decode* them into tokenized output. This is commonly achieved in a multi-modal vision-language framework that incorporates the language modality, as exemplified by either projecting visual tokens to LLMs via linear layers (Koh et al., 2023; Lin et al., 2023; Merullo et al., 2023; Schwettmann et al., 2023) or employing cross-attention mechanisms between visual and language tokens (Alayrac et al., 2022; Li et al., 2022; 2023; Wang et al., 2023). As we explore the limits of utilizing LLMs for computer vision tasks, an interesting question arises: can LLMs effectively handle tasks that are *exclusively* visual, without any reliance on language?

This paper provides a positive demonstration of feasibility in addressing this question, by introducing a straightforward yet previously overlooked approach: incorporating a *frozen* transformer block from a *pre-trained* LLM as a general-purpose visual *encoder* layer, directly processing the visual tokens. Specifically, as illustrated in Fig. 1a and Fig. 1b, our design involves the following steps: (1) extract a *frozen* LLM transformer block and append it on top of the original visual encoder; (2) insert *trainable* linear layers before and after the added LLM block to align the feature dimensions; and (3) *freeze* the LLM transformer while optimizing the other modules as usual during training.

Surprisingly, this simple design enhances performance across *a wide spectrum of tasks*, including 2D and 3D recognition (image and point cloud classification), video understanding (action recognition), and non-semantic (motion forecasting) tasks. In addition to these purely visual tasks, our approach

---

\*Equal contribution.

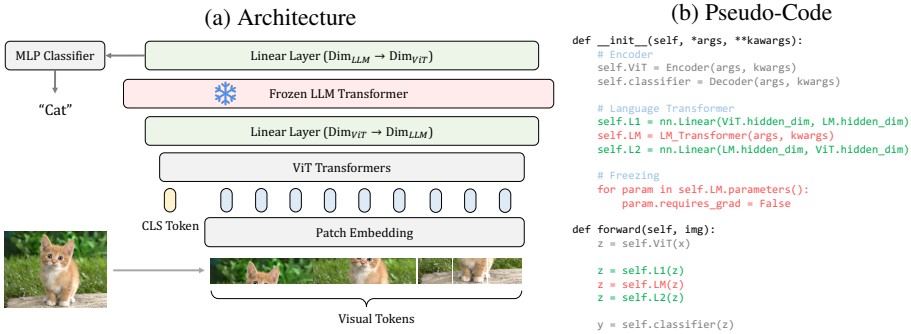

Figure 1: Our straightforward method of using a *frozen* transformer block from *pre-trained* LLMs as a visual encoder layer. Visualized with an example of ViT (Dosovitskiy et al., 2021). **(a)** Our design simply appends a frozen transformer block (pink) on top of the regular visual encoder (gray). Only two trainable linear layers (green) are added to align the feature dimensions. **(b)** Pytorch-style pseudo-code shows the simplicity of our approach.

is also effective in multi-modal tasks (2D/3D visual question answering and image-text retrieval). Notably, such improvements are general across various types of LLMs like LLaMA (Touvron et al., 2023) and OPT (Zhang et al., 2022), as well as different LLM transformer blocks.

Our discovery of using a pre-trained LLM transformer block as a visual encoder layer is intriguing, because it significantly deviates from the conventional designs of vision-language models (VLMs). In particular, our treatment of LLM transformers as *encoders* (1) operates independently of language prompts, inputs, or outputs; (2) allows for training from scratch without the need for pre-trained backbones like CLIP (Radford et al., 2021); and (3) decouples and simplifies the usage of LLMs into separate transformer blocks.

However, one crucial question remains: *why are LLMs effective in visual encoding, given that they have been exclusively trained on text and have never encountered visual input?* To this end, we propose the *information filtering* hypothesis: the pre-trained LLM transformer blocks discern informative visual tokens and further amplify their contribution to the latent representation. This hypothesis stems from our observation across multiple tasks, where the feature activation consistently exhibits a stronger focus on relevant regions, after integrating the frozen LLM transformer blocks.

In summary, we have made the following contributions:

- We discover that *using a frozen transformer block from pre-trained LLMs as a visual encoder layer* facilitates a diverse range of tasks, by introducing a simple yet under-explored approach.
- We demonstrate that the benefits of frozen LLM transformers in visual encoding are a general phenomenon, through our investigation on various LLMs and transformer blocks.
- We propose the *information filtering hypothesis* to explain the effectiveness of frozen LLM transformers in processing visual tokens: the incorporated LLM blocks distinguish the informative tokens and amplify their effect.

We hope that our work will drawn attention to the intriguing application of employing LLM transformers as versatile encoders, not only for visual inputs but potentially also for other modalities. Additionally, we hope to inspire new perspectives on understanding LLMs and VLMs.

## 2 RELATED WORK

**Large language models.** Pre-training transformers (Vaswani et al., 2017) with *masked token prediction* facilitates the generalizability of language models (LMs) to various tasks, represented by BERT (Kenton & Toutanova, 2019). Later on, larger models at scale are proposed guided by the scaling law (Kaplan et al., 2020), such as GPT (Brown et al., 2020), LLaMA (Touvron et al., 2023), OPT (Zhang et al., 2022), *etc*. These large models with tens of billions of parameters unlock the intriguing ability of in-context learning and excellent zero-shot performance on various tasks. Our work highlights the interesting discovery that the transformer blocks in such large language models (LLMs) are able to interact with visual data and enhance a wide spectrum of computer vision tasks.

**Language models for visual tasks.** LMs are mostly used as *text encoders* for vision-language models (VLMs) (Dou et al., 2022; Kim et al., 2021) or image-text pre-training (Radford et al., 2021) before the emergence of LLMs. After the creation of LLMs, their code generation ability

encourages flexibly combining the vision algorithms for user queries, represented by visual programming (Gupta & Kembhavi, 2023). In addition, the strong ability of LLMs has also elicited using them as generalizable *decoders*, *i.e.*, translating the latent feature into output tokens. These frameworks commonly project visual features to the input layer of LLMs directly (Guo et al., 2023; Koh et al., 2023; Lin et al., 2023; Merullo et al., 2023; Chen et al., 2023) or use the structures of the latent bottleneck (Jaegle et al., 2021) to further encode visual tokens (Alayrac et al., 2022; Hong et al., 2023; Li et al., 2022; 2023; Wang et al., 2023). Our exploration reveals the potential of considering the transformer blocks in LLMs as general-purpose *encoders* for *visual* data, as opposed to the previous usages of either pure *encoders* for text embeddings or *decoders* for tokenized outputs.

**Interpreting neural networks.** Understanding neural networks begins by visualizing the convolutional patterns in low-level layers (Erhan et al., 2009). With a deeper interest in semantics, attribution-based methods like Grad-CAM (Selvaraju et al., 2017) further analyze the contribution of neurons for a certain class. Network dissection (Bau et al., 2017; Zhou et al., 2018) also discovers that neural network units correspond to semantic concepts. For LLMs, researchers find that the knowledge is mainly located at the linear layers in feedforward networks (FFN) (Dai et al., 2022; Geva et al., 2020; Meng et al., 2022a;b), and corresponds to visual concepts (Schwettmann et al., 2023). Compared with them, we study a new scenario of why a pre-trained LLM transformer can benefit visual encoding and propose the *information filtering hypothesis*.

## 3 METHOD: FROZEN LLM TRANSFORMERS FOR VISUAL ENCODING

**Framework design.** We formally introduce using a pre-trained LLM transformer as a visual encoder layer shown in Fig. 1a. Without loss of generality, we consider a neural network that maps input $x$ to latent representation $z$ and predicts labels $y$ with an encoder $\mathbf{F}_E$ and a decoder $\mathbf{F}_D$,

$$\mathbf{F}_E(x) \to z, \quad \mathbf{F}_D(z) \to y. \tag{1}$$

Then a *single pre-trained* transformer block from an LLM like LLaMA (Touvron et al., 2023), denoted as $\mathbf{F}_{LM}$, is inserted between the encoder $\mathbf{F}_E$ and decoder $\mathbf{F}_D$. As the feature dimensions are different between the encoder $\mathbf{F}_E$ and the language transformer $\mathbf{F}_{LM}$, we employ two linear layers $\mathbf{F}_L^1$ and $\mathbf{F}_L^2$ before and after $\mathbf{F}_{LM}$ to align the dimensionality. These modify the neural network into

$$\mathbf{F}_E(x) \to z, \quad \mathbf{F}_L^2 \cdot \mathbf{F}_{LM} \cdot \mathbf{F}_L^1(z) \to z', \quad \mathbf{F}_D(z') \to y. \tag{2}$$

In the training stage, the pre-trained transformer $\mathbf{F}_{LM}$ remains *frozen*, as in the pseudo-code of Fig. 1b, while all the other modules are trained normally, including $\mathbf{F}_L^1$ and $\mathbf{F}_L^2$.

**Comparison with vision-language models.** Our approach appears similar to recent vision-language models (VLMs) at the first glance, such as Lin et al. (2023), FROMAGe (Koh et al., 2023), and LiMBeR (Merullo et al., 2023), where linear layers directly project visual features to the input space of LLMs. However, our approach is different, because the linear layer $\mathbf{F}_L^1$ does not necessarily align the visual representation $z$ into the language space. Concretely, this is reflected in three aspects: (1) **Independence of visual pre-training.** Our paradigm supports training-from-scratch without relying on pre-trained visual encoders like CLIP (Radford et al., 2021). (2) **Independence of language.** Our framework can function without language-based input or prompts, and it is applicable for general visual representation learning instead of only vision-language tasks. (3) **Independence of transformer blocks.** Previous VLMs treat an entire LLM as a coherent module, while our framework separates each transformer block as an independent layer for visual encoding.

**Comparison with LLMs.** We substantially change the behaviors of LLM transformers, due to the distinct formats between visual and text data. (1) **Attention mask.** LLMs commonly utilize auto-regressive masks to mimic the order of text generation. However, the tokens in visual data come all at once, such as the image tokens of the cat (Fig. 1a). So we abandon auto-regressive attention masks and only use attention masks to indicate the padded tokens. (2) **Positional embedding.** The positional embedding in LLMs, *e.g.*, rotary positional embedding (Su et al., 2021) in LLaMA, is not a common option for visual encoders. Therefore, we remove the positional embeddings of LLMs for simplicity and consistency with the original visual backbones. Considering the importance of attention masks and positional embeddings in transformers, it is surprising in hindsight that our framework has a positive influence on visual tasks.

## 4 APPLICABILITY OF LLM TRANSFORMERS FOR VISUAL TASKS

We instantiate our framework to various tasks and observe the wide applicability of pre-trained LLM transformers. Our experiments cover 2D (image classification) and 3D (point cloud classification),

| Model | ImageNet | ImageNet-C | ImageNet-A | ImageNet-SK | ImageNet-R |
|---|---|---|---|---|---|
| ViT-T | 72.1 | 43.9 | 7.7 | 19.6 | 32.3 |
| +LLaMA | **73.2** | **45.8** | **8.7** | **20.6** | **33.8** |
| ViT-S | 80.1 | 57.2 | 20.5 | 28.9 | 42.1 |
| +LLaMA | **80.7** | **58.7** | **22.7** | **30.5** | **42.8** |
| ViT-B | 80.6 | 60.5 | 23.4 | 31.9 | 43.5 |
| +LLaMA | **81.7** | **62.1** | **26.9** | **33.2** | **44.3** |

Table 1: Incorporating a single transformer block from LLaMA to ViT models consistently improves both accuracy (ImageNet) and robustness (ImageNet-{C,A,SK,R}).

single-frame and multi-frame (action recognition), semantics and motion (motion forecasting), and tasks involving linguistic input or not (2D and 3D vision-language tasks). By default, we adopt the last transformer block from LLaMA-7B (Touvron et al., 2023). Our framework achieves *consistent* and *significant* improvements across these tasks following the standards of prior work. More analysis on our designs is in Sec. 5.

## 4.1 IMAGE CLASSIFICATION

Image classification is the most common challenge for representation learning. We conduct experiments on ImageNet1k (Deng et al., 2009), and additionally evaluate on robustness benchmarks: corrupted images from ImageNet-C (Hendrycks & Dietterich, 2018), natural adversarial images from ImageNet-A (Hendrycks et al., 2021b), and out-of-distribution images from ImageNet-SK (Wang et al., 2019) and ImageNet-R (Hendrycks et al., 2021a).

Without loss of generality, we select ViT (Dosovitskiy et al., 2021) due to its wide use and native support for transformers. Following the notation in Eqn. 2, the encoder $\mathbf{F}_E$ is the set of self-attention transformer blocks and the decoder $\mathbf{F}_D$ denotes a linear classifier. An intuitive illustration is in Fig. 1a. We train both the baseline ViT models and ViT+LLaMA from scratch following the same configuration of DeiT (Touvron et al., 2021). More details are in Sec. C.1.

The accuracy of ViT models consistently improves after incorporating the frozen LLaMA transformer block as in Table 1, including both the accuracy on clean ImageNet images and the robustness on corrupted or adversarial images. Our further experiments validate that the improvement is closely related to the LLM transformer instead of the sole consequence of an increased model capacity. Please refer to Sec. 5.1 and Sec. B.2 for details.

## 4.2 POINT CLOUD CLASSIFICATION

Point cloud classification handles a fundamentally different modality compared with images. The models predict labels by processing unordered 3D points and understanding the geometry. Our experiments cover two common datasets: ScanObjectNN (Uy et al., 2019) and ModelNet40 (Goyal et al., 2021). ScanObjectNN contains three splits: background (BG), foreground (OBJ), and clipped (T50) points. For ModelNet40, we experiment with different densities (1k, 4k, 8k) of points.

| Model | ScanObjectNN | | |
|---|---|---|---|
| | BG | OBJ | T50 |
| Point-BERT | 87.4 | 88.1 | 83.1 |
| +LLaMA | **88.0** | **88.5** | **83.8** |

| Model | ModelNet40 | | |
|---|---|---|---|
| | 1k | 4k | 8k |
| Point-BERT | **92.67** | **92.91** | 93.19 |
| +LLaMA | 92.42 | 92.82 | **93.56** |

Table 2: LLM transformer improves point cloud classification.

We adopt Point-BERT (Yu et al., 2021) and load its pre-trained parameters on ShapeNet (Chang et al., 2015). Then we append the LLaMA transformer after its final attention block before fine-tuning on point cloud classification datasets. Details are in Sec. C.2.

As shown in Table 2, our approach improves the accuracy for point cloud classification, further supporting the applicability of using a frozen LLM transformer as a visual encoding layer. Note that the accuracy slightly drops on ModelNet40 with 1k and 4k points, due to the saturation and $\sim$0.2% variance of performance on ModelNet40 which is also analyzed in Ma et al. (2022). However, with an increased number of points (8k), the improvement of the LLaMA transformer is noticeable on ModelNet40. More importantly, on the more challenging ScanObjectNN, our approach improves the accuracy consistently and significantly. This experiment also shows that our framework is compatible with fine-tuning setups, in addition to training-from-scratch scenarios in Sec. 4.1.

## 4.3 ACTION RECOGNITION

For the video modality, we apply the pre-trained LLM transformer block to action recognition, where the algorithm predicts the action labels of video clips. We choose the benchmark of "Something-

| (a) 2D VQA and Image Retrieval | | | | | | |
|---|---|---|---|---|---|---|
| Model | VQAv2 (Tes-dev) | | | Flickr30k (Val) | | |
| | Overall | Yes/No | Number | Other | IR@1 | IR@5 | IR@10 |
| METER | 69.60 | 85.08 | 47.82 | 61.37 | 49.66 | 80.86 | 89.48 |
| + LLaMA | **70.23** | **85.70** | **48.98** | **61.89** | **50.22** | **82.26** | **90.08** |

| (b) 3D VQA | | |
|---|---|---|
| Methods | EM@1 | EM@10 |
| ScanQA | 46.58 | 85.97 |
| SQA3D | 47.20 | 86.82 |
| SQA3D-LLaMA | **48.09** | **89.03** |

Table 5: Frozen LLaMA transformer enhances both 2D **(a)** and 3D **(b)** vision-language models.

something-v2" (SSv2) (Goyal et al., 2017b) for evaluation, because it highlights the challenge of understanding cross-frame movement, instead of relying on single-frame semantics.

We follow VideoMAE (Tong et al., 2022) and adopt the simple yet effective ViT backbones. Different from patches of tokens in 2D images, the video tokens are cubes spanning both spatially and temporally. Identical to Fig. 1a, we place the LLaMA transformer behind the last self-attention block in ViT. Our training setup also adopts the two-step practice in VideoMAE: (1) initialize ViT transformers from MAE (He et al., 2022) pre-training; (2) add the LLM transformer and then fine-tune on the SSv2 dataset using the same configuration of VideoMAE. More details are in Sec. C.3.

| Model | Acc1 | Acc5 |
|---|---|---|
| ViT-S | 64.71 | 89.15 |
| +LLaMA | **65.89** | **89.93** |
| ViT-B | 64.97 | 89.50 |
| +LLaMA | **66.03** | **90.25** |

Table 3: Pre-trained LLM transformer improves action recognition on SSv2.

The LLaMA transformer enhances the accuracy for both ViT-S and ViT-B in Table 3, supporting the applicability of our framework for videos. To clarify, our baseline accuracy is lower than that reported in VideoMAE's original paper, because VideoMAE used 32/64 GPUs to enable a larger batch size than our computational resources. Nonetheless, we control the settings identical between ViT and ViT-LLaMA for a fair comparison and indicate the positive effects of using LLM transformers.

## 4.4 MOTION FORECASTING

We select motion forecasting as an example of a non-semantic task. It is safety-critical for autonomous driving and capitalizes on the understanding of dynamics, agent-agent interaction, and agent-lane relationship. The input commonly includes the historical trajectories of agents and way-points of lane segments, which are both represented in polylines on the bird's-eye view (BEV). The desired output is a set of $K$ most possible future trajectories.

| Model | ADE↓ | FDE↓ | MR↓ |
|---|---|---|---|
| VectorNet | 0.77 | 1.23 | 13.2 |
| +LLaMA | **0.76** | **1.20** | **12.7** |
| mmTransformer | 0.72 | 1.10 | 10.7 |
| +LLaMA | **0.71** | **1.08** | **10.5** |

Table 4: LLM transformer layer is beneficial for motion forecasting.

We conduct experiments on Argoverse (Chang et al., 2019). The evaluation metrics are minimum average displacement (ADE), minimum final displacement (FDE), and miss rate (MR), which calculate the errors of predictions from different aspects and are better at lower values. We apply the frozen LLM transformer to VectorNet (Gao et al., 2020) and mmTransformer (Liu et al., 2021). They first convert the agents and lanes into features, and then our LLaMA transformer block processes these agent and lane tokens. Demonstration and details are in Sec. C.4.

According to Table 4, the models with LLaMA forecast better trajectories. However, we notice that the improvement is less significant compared with semantic tasks, which reflects the preference of LLM transformers for encoding rich semantics over object movements.

## 4.5 VISION-LANGUAGE TASKS

**2D vision-language tasks.** The benefits of frozen LLM transformers for visual encoding are not limited to purely visual tasks. We experiment with 2D vision-language (VL) tasks, including visual question answering (VQA) on VQAv2 (Goyal et al., 2017c) and zero-shot image retrieval on Flickr30k (Plummer et al., 2015). We adopt the widely-used METER (Dou et al., 2022) as our baseline. It extracts uni-modal features for images and text, fuses cross-modal features, and decodes the output from the cross-modal features. We insert the LLM transformer block after the cross-modal fusion. An intuitive illustration is in Fig. H. During training, our setup follows Shi et al. (2023): initialize the image encoder with CLIP-B/32 (Radford et al., 2021) and the text encoder with RoBERTa (Liu et al., 2019), and then fine-tune on VQAv2 or Flickr30k. Details are in Sec. C.5. As in Table 5a, both of the 2D VL tasks are significantly enhanced with the LLaMA transformer. This evidence supports the potential of a frozen LLM transformer for multi-modal tasks.

**3D visual question answering.** We extend our proposed idea into 3D VQA, which requires comprehending an input 3D scene represented by a point cloud or multi-view images and then answering questions. 3D VQA challenges the ability to ground language in 3D context. We conduct our exper-

iments on the SQA3D (Ma et al., 2023) dataset, comparing with baseline methods and state of the arts (Ma et al., 2023; Azuma et al., 2022) on the exact match (EM) metric. We follow the baseline SQA3D to process the textual input with LSTM (Hochreiter & Schmidhuber, 1997) and 3D point clouds with VoteNet (Qi et al., 2019). Here, we add the LLM block after the VL fusion, which is consistent with our 2D VL design (Sec. 4.5). More details are in Sec. C.6. According to Table 5b, adding a frozen LLM transformer effectively enhances the QA ability of models. The full comparison with detailed breakdown metrics is in Table E.

## 5 ANALYSIS ON LLM TRANSFORMERS FOR VISUAL TASKS

We justify our design choices (Sec. 5.1) and illustrate the wide applicability of our framework to various LLMs and transformer layers (Sec. 5.2). Our investigation also discovers that sufficiently large LLMs are the premise of benefiting visual encoding with a frozen transformer (Sec. B.3) and discusses the place to insert LLM blocks (Sec. B.4).

### 5.1 ABLATION STUDY ON DESIGN CHOICES

**Model capacity.** Regarding the wide applicability of frozen LLM transformers, we question if the improvement mainly comes from the increased capacity of the linear layers $\mathbf{F}_L^1$ and $\mathbf{F}_L^2$, instead of the pre-trained weights in LLM transformers $\mathbf{F}_{LM}$. To analyze model capacity, we create ViT-S-MLP, which has identical trainable parameters compared with ViT-S-LLaMA. Concretely, ViT-S-MLP removes the LLM block $\mathbf{F}_{LM}$, and then inserts a GeLU activation (Hendrycks & Gimpel, 2016) and layer normalization (Ba et al., 2016) between $\mathbf{F}_L^1$ and $\mathbf{F}_L^2$. It also adopts the identical train-

| Model | Acc |
|---|---|
| ViT-S | 80.1 |
| ViT-S-LLaMA | **80.7** |
| ViT-S-MLP | 80.4 |
| ViT-S-LLaMA-FT | 78.9 |

Table 6: Ablation study on model capacity and fine-tuning.

ing procedure as ViT and ViT-LLaMA in Sec. 4.1 for a fair comparison. The results are summarized in Table 6: the ViT-S-MLP has better performance than ViT-S due to its increased capacity, but the improvement is only about half of ViT-S-LLaMA. Therefore, the LLM transformer weights are crucial for the improvement and the observed benefits are *not* mere consequences of increased model capacity. Investigation with more tasks and baselines are in Sec. B.2.

**Fine-tuning.** We further verify whether fine-tuning the language transformer (ViT-S-LLaMA-FT) is better than freezing it. As in Table 6, fine-tuning decreases the performance compared with ViT-S-LLaMA. We analyze this phenomenon by visualizing the loss curves in Fig. F and training under an additional setting of 100 epochs. Although fine-tuning improves performance under short training (100 epochs in Table A), it hurts the accuracy when trained sufficiently: in Fig. F, ViT-S-LLaMA-FT shows lower training loss but relatively larger validation loss, which indicates overfitting. Thus, our observation demonstrates the challenges of training large transformers, and we accordingly freeze the LLM transformers in our design because of its simplicity and effectiveness.

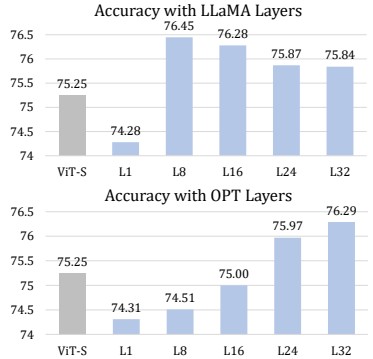

Figure 2: Various LLM transformer layers improve the accuracy.

### 5.2 VARYING LLM TRANSFORMER LAYERS

We discover that different LLM transformers influence visual representation learning significantly within our framework, even though they have identical capacity. Specifically, we use transformer blocks from diverse depths of LLaMA-7B (Touvron et al., 2023) and OPT (Zhang et al., 2022) onto ViT-S. The models are trained in the ablation study setting of 100 epochs. More details are in Sec. C.7. As shown in Fig. 2, the layer type significantly changes the performance. These experiments also validate that *our framework is applicable to various LLMs and transformer layers*, and highlight the importance of selecting proper transformer layers. We additionally observe that the last LLM layers consistently improve the performance although they might not be optimal.

## 6 INFORMATION FILTERING HYPOTHESIS

This section aims to explain how a pre-trained and frozen LLM transformer benefits visual tasks. Intuitively, our hypothesis can be stated as:

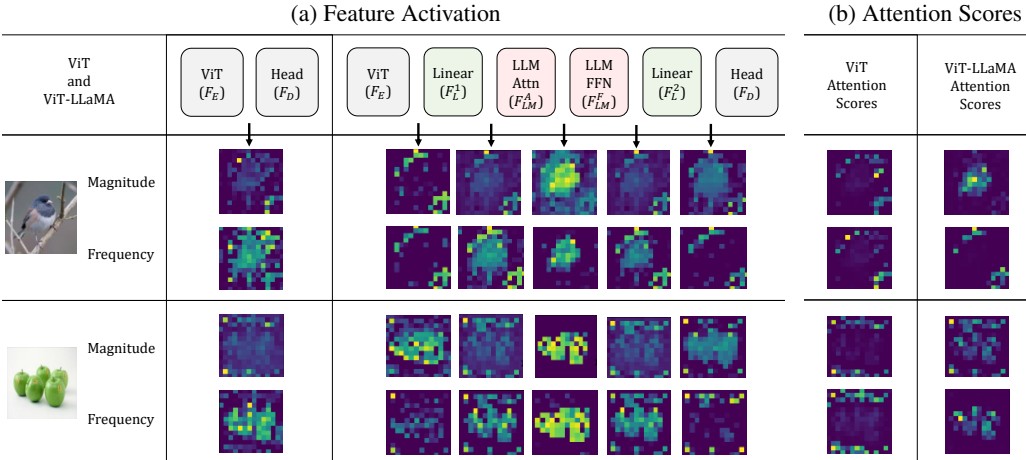

Figure 3: **(a)** Feature activation regarding both magnitudes and frequencies of features. We highlight that ViT-LLaMA demonstrates the emergent tendency of object segmentation compared with ViT, indicating its ability to select informative tokens. **(b)** Attention scores between the CLS and visual tokens. The attention from ViT is commonly noisy (left). Though ViT-LLaMA improves the concentration on a few heads, most of the attention heads are still noisy. Both good and bad attention from ViT-LLaMA are sampled for demonstration purpose.

**Information filtering hypothesis.** A pre-trained LLM transformer functions as a "filter" that *distinguishes the informative tokens* and *amplifies their contribution for the prediction*, in the form of enlarged magnitudes or frequencies in the feature activation.

We first derive this hypothesis in the context of image classification (Sec. 6.1), then provide quantitative investigation (Sec. 6.2), and discuss the observation on other tasks (Sec. 6.3). Due to space limits, we include more details in Sec. A and discuss limitations in Sec. A.4.

## 6.1 QUALITATIVE DERIVATION OF INFORMATION FILTERING HYPOTHESIS

**Emergent concentration on informative tokens.** Our hypothesis originates from the emergent behavior that *the feature activation highlights the informative tokens* after adding a pre-trained LLM transformer. In the analysis, we extract the activation of features after each layer as Fig. 3a, including the original ViT $\mathbf{F}_E$, the attention layer $\mathbf{F}_{LM}^A$ and feedforward network $\mathbf{F}_{LM}^F$ in the LLM transformer, and the linear layers $\mathbf{F}_L^1$ and $\mathbf{F}_L^2$. Notably, the feature activation is calculated regarding both *magnitudes* (L2-norm after centering) and *frequencies* (L2-norm of angles after Fourier transformation)[1]. The different layers in Fig. 3a indeed show diverse preferences over magnitudes or frequencies.

As clearly demonstrated in Fig. 3a, the token activation better captures the regions of target objects after adding the LLM transformer, especially the magnitudes of $\mathbf{F}_L^2$ and frequencies of $\mathbf{F}_{LM}^A$. Their tendency of segmentation is a surprising discovery, because emergent segmentation is *only* observed in self-supervised learned (Caron et al., 2021) or specially-designed (Darcet et al., 2024; Shi et al., 2023; Yang et al., 2024) ViTs. More importantly, the activation's concentration on the target object directly supports our hypothesis as evidence of selecting the informative tokens.

**Noisy attention scores.** In contrast to the feature activation, attention scores struggle to capture the relevant visual tokens for prediction. We investigate the attention scores between the CLS token and visual tokens in the last transformer block, which are the last self-attention block in $\mathbf{F}_E$ for ViT and the transformer $\mathbf{F}_{LM}$ for ViT-LLaMA, respectively. Ideal attention scores that distinguish the target object should exhibit object segmentation patterns like DINO (Caron et al., 2021). However, supervised ViT models commonly have noisy attention scores (left part in Fig. 3b). Although ViT-LLaMA illustrates the ability of emergent segmentation in a few attention heads, most of the attention scores also suffer from scattering and noisiness. These observations contrast the feature activation and indicate that the benefits of LLM transformers cannot be simply attributed to attention scores, since attention scores fail to reliably contribute correct visual tokens.

**Deriving the amplification of informative tokens.** According to our visualization in Fig. 3a, the frozen LLM transformer distinguishes the informative tokens. Intuitively, such tokens naturally

---

[1]Details of calculation are in Sec. A.5.

benefit the downstream decoding, but this is straightforward *only* when the decoder directly takes the visual tokens as input. As a counterexample, ViT utilizes a CLS token for classification, and the visual tokens output by $\mathbf{F}_L^2$ is not the input to the decoder and always receives zero gradients during training. To bridge the gap in CLS token scenarios, the second half of our hypothesis is necessary: the frozen LLM transformer *amplifies* the contribution of informative tokens.

Concretely, the calculation of the CLS token is,

$$z_L^2[\text{CLS}] = \mathbf{F}_L^2 \cdot \mathbf{F}_{LM}^F \cdot \mathbf{F}_{LM}^A \left( \sum_{v \in V} w_v \, z_L^1[v] \right), \tag{3}$$

where $V$ denotes visual tokens, and $w_v$ denotes the weight of visual token $v$. Eqn. 3 describes the process of (1) aggregating visual tokens $z_L^1[v]$ guided by the attention scores $w_v$; and (2) sequentially flowing through the subsequent layers, including the LLM transformer's attention head $\mathbf{F}_{LM}^A$, feed-forward network $\mathbf{F}_{LM}^F$, and the second linear layer $\mathbf{F}_L^2$. To concentrate on explaining visual tokens, Eqn. 3 also slightly simplifies self-attention by removing the CLS token from the right-hand side.

In Eqn. 3, the useful visual tokens are not reliably attributed to the CLS token, because the attention scores $w_v$ are observed to be noisy. Therefore, our objective is to connect the visual tokens in Eqn. 4,

$$z_L^2[v] = \mathbf{F}_L^2 \cdot \mathbf{F}_{LM}^F \cdot \mathbf{F}_{LM}^A(z_L^1[v]), \text{ where } v \in V, \tag{4}$$

which are informative (as in Fig. 3a), to the final feature representation $z_L^2[\text{CLS}]$. This pursuit makes us notice that attention scores $w_v$ are noisy while token features $z_L^2[v]$ are informative. Such a contradiction indicates that *amplification of the informative tokens* is a more plausible explanation, compared with better attention scores. We express the hypothesis below in a formal way, which is a simple change to Eqn. 3:

$$z_L^2[\text{CLS}] \quad \propto \quad \sum_{v \in V} w_v \underbrace{\left( \mathbf{F}_L^2 \cdot \mathbf{F}_{LM}^F \cdot \mathbf{F}_{LM}^A(z_L^1[v]) \right)}_{z_L^2[v]}. \quad [\text{Hypothesis}] \tag{5}$$

Eqn. 5 holds equal under the special case of $\mathbf{F}_L^2 \cdot \mathbf{F}_{LM}^F \cdot \mathbf{F}_{LM}^A$ being linear transformation. This equation explains how the informative tokens in Fig. 3a are *implicitly* supervised in the CLS token.

As a brief remark, our derivation builds upon two observed pieces of evidence: (1) visual tokens concentrating on informative regions; and (2) noisy attention scores. These lead to the first half of our hypothesis and explain the benefits when visual tokens are direct input to decoders. By further connecting visual tokens to the CLS token, we propose that the inserted LLM transformer amplifies the effects of informative tokens. A more thorough version of the derivation is in Sec. A.1.

## 6.2 QUANTITATIVE EVIDENCE

The qualitative observation in Sec. 6 is further supported with quantitative evidence. Specifically, we use the ImageNet-S (Gao et al., 2022) dataset to provide the ground truth of "informative regions" from its annotation of semantic segmentation masks. To assess the fidelity of feature activation and attention scores, we first generate pseudo-masks highlighting their concentrating regions, *i.e.*, the tokens with larger activation or attention scores than the others on the same image. Then the quality of features and attention scores are reflected by the mIoU (mean intersection-over-union) between their pseudo-masks and ground truth segmentation masks. Implementation details are in Sec. A.3.

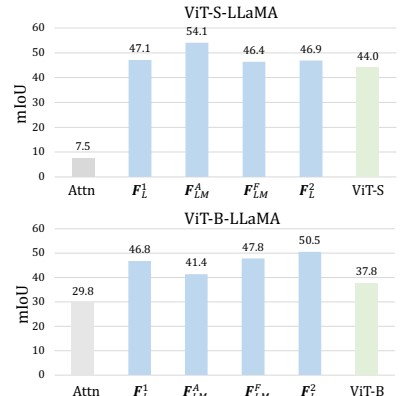

Figure 4: Pseudo-masks from ViT-LLaMA's features ($\mathbf{F}_L^2$) have larger mIoU than attention scores and ViT.

Finally, we summarize the mIoU statistics for feature activation and attention scores in Fig. 4. As demonstrated, both ViT-S-LLaMA and ViT-B-LLaMA have better mIoU of pseudo-masks than attention scores. This directly supports our hypothesis that the features $\{z_L^2[v]) | v \in V\}$ contribute more reliably than attention scores $\{w_v | v \in V\}$. We additionally notice that the pseudo-masks from ViT-LLaMA generally have larger mIoU compared with ViT, which reflects the benefits of training ViTs with a frozen LLM

Figure 5: Token activation in action recognition. Video tokens are activated jointly in all the frames, and every video token is a cube with shape $2 \times 16 \times 16$. After adding the LLM transformer, the model better concentrates on the relevant objects and hands ("low threshold") and more accurately focuses on frames with hand-object interaction ("high threshold").

transformer. The advantage of the feature in the first linear layer $\mathbf{F}_L^1$ also reveals that training with our framework is beneficial to even earlier stages of features. However, we would like to clarify that the pseudo-masks from either magnitude or frequency activation are intuitive but lossy measures to quantify feature quality, because neural networks can encode information in other formats. Therefore, better measurements to analyze network layers will be meaningful for future work.

### 6.3 INFORMATION FILTERING HYPOTHESIS ON OTHER TASKS

The previous sections mainly discuss our information filtering hypothesis in terms of image classification. Meanwhile, we also discover supportive evidence of our hypothesis on various other tasks. This section investigates *action recognition* as an example, and Sec. A.2 covers additional tasks including point cloud classification, 2D VQA, and 3D VQA.

We mainly analyze the information filtering hypothesis in action recognition *qualitatively*, because the ground truth of "relevant regions" is difficult to quantify for this task. In practice, we follow a similar procedure in Sec. 6.1 and visualize the activation of video tokens in Fig. 5, and display the most highly activated tokens according to either low or high thresholds for better clarity. At a low activation threshold, we notice that the video tokens from ViT-S-LLaMA better capture the foreground areas of hands and manipulated objects than ViT-S. With the video tokens in VideoMAE activated both spatially and temporally, we further increase the threshold to demonstrate its ability to select informative frames. As in the "high threshold" row of Fig. 5, ViT-S-LLaMA more accurately focuses on the middle frames with actual human-object interaction. Therefore, we conclude that the informative video tokens are indeed distinguished and augmented in action recognition, which aligns with the information filtering hypothesis.

## 7 CONCLUSION

In this work, we explore the unexpected capability of large language models (LLMs) as encoders for visual tasks, a significant departure from their conventional text-based applications. By seamlessly integrating a frozen transformer block from pre-trained LLMs into visual encoders, we observe consistent performance enhancements across diverse visual challenges, including 2D image and video classification, 3D point cloud classification, motion forecasting, and 2D and 3D vision-language tasks. This phenomenon, underpinned by our proposed information filtering hypothesis, highlights the inherent adaptability and versatility of LLMs for more general representation learning. We hope that our insights will catalyze further exploration into the uncharted fields of LLM applications and foster innovative strategies to harness their potential in novel ways.

**Discussion and limitations.** We have validated the capability of pre-trained, frozen language transformers across a wide spectrum of visual tasks. It is important to note that our goal is to methodically explore this under-investigated problem. Therefore, our experiments are designed to maximize the diversity of tasks under fair comparisons with well-established or competitive baselines, rather than striving for state-of-the-art performance for all tasks, which is also constrained by our computational resources. We leave scaling up the experiments to state-of-the-art levels for all tasks as interesting future work. Meanwhile, we also notice that our information filtering hypothesis has not covered several intriguing questions, *e.g.*, how to quantify the functions of different layers and analyze how the training process facilitates the visual token features to cooperate with the language transformer, which are also meaningful directions.

**Acknowledgement.** We thank Baifeng Shi and Shoufa Chen for their valuable discussion and suggestions on implementation and model training. This work was supported in part by NSF Grant 2106825, NIFA Award 2020-67021-32799, the Jump ARCHES endowment through the Health Care Engineering Systems Center, and the IBM-Illinois Discovery Accelerator Institute. This work used NVIDIA GPUs at NCSA Delta through allocation CIS220014 and CIS230012 from the Advanced Cyberinfrastructure Coordination Ecosystem: Services & Support (ACCESS) program, which is supported by NSF Grants #2138259, #2138286, #2138307, #2137603, and #2138296.

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

# A   THOROUGH DISCUSSION ON THE INFORMATION FILTERING HYPOTHESIS

## A.1   DETAILED DERIVATION OF THE HYPOTHESIS

We expand the details of several steps in our derivation (Sec. 6.1) for better clarity. Beginning from the formula of the CLS token below,

$$z_L^2[\text{CLS}] = \mathbf{F}_L^2 \cdot \mathbf{F}_{LM}^F \cdot \mathbf{F}_{LM}^A \left( \sum_{v \in V} w_v \, z_L^1[v] \right), \tag{A}$$

which is identical to Eqn. 3 in the main paper, we further separate the visual tokens into the subset of informative tokens $V_i$ and uninformative tokens $V_u$. Intuitively, the tokens corresponding to the foreground objects are informative, and the background ones are uninformative. This changes Eqn. A into,

$$z_L^2[\text{CLS}] = \mathbf{F}_L^2 \cdot \mathbf{F}_{LM}^F \cdot \mathbf{F}_{LM}^A \left( \sum_{i \in V_i} w_i \, z_L^1[i] + \sum_{u \in V_u} w_u \, z_L^1[u] \right). \tag{B}$$

Using the same notation, our observation on the feature activation can be stated as: the attention weights for informative tokens $\{w_i, i \in V_i\}$ are still noisy after incorporating the frozen LLM transformer, while the final visual tokens shown as below have emergent concentration on target regions:

$$z_L^2[i] = \mathbf{F}_L^2 \cdot \mathbf{F}_{LM}^F \cdot \mathbf{F}_{LM}^A(z_L^1[i]), \text{ where } i \in V_i. \tag{C}$$

Combining Eqn. B and Eqn. C inspires us to express our hypothesis in terms of the connection between visual and CLS tokens with Eqn. D below, where the added modules $\mathbf{F}_L^2 \cdot \mathbf{F}_{LM}^F \cdot \mathbf{F}_{LM}^A$ augment the informative tokens $V_i$ and lead to better prediction:

$$z_L^2[\text{CLS}] \quad \propto \quad \sum_{i \in V_i} w_i \underbrace{\left( \mathbf{F}_L^2 \cdot \mathbf{F}_{LM}^F \cdot \mathbf{F}_{LM}^A(z_L^1[i]) \right)}_{z_L^2[i]} + \sum_{u \in V_u} w_u \underbrace{\left( \mathbf{F}_L^2 \cdot \mathbf{F}_{LM}^F \cdot \mathbf{F}_{LM}^A(z_L^1[u]) \right)}_{z_L^2[u]}. \quad [\text{Hypothesis}] \tag{D}$$

This is a more thorough expression of our hypothesis in the main paper (Eqn. 5) to better differentiate the role of informative tokens in our hypothesis.

## A.2   INFORMATION FILTERING HYPOTHESIS ON OTHER TASKS

This section provides the evidence for our information filtering hypothesis in other tasks, supplementing the discussion on image classification (Sec. 6.1) and action recognition (Sec. 6.3). Specifically, we observe that the frozen LLM transformer selects and amplifies information tokens in point cloud classification, 2D visual question answering (VQA), and 3D VQA. The tasks of motion forecasting and image-text retrieval are not illustrated, because it is more abstract to intuitively define their "informative" tokens. For the investigated tasks, we mainly analyze qualitatively because the ground truth for relevant regions is ambiguous on such tasks, unlike the segmentation masks for image classification (Sec. 6.2).

**Point cloud classification.**   We visualize the activation of the point tokens in point cloud classification before and after adding the frozen LLM transformer in Fig. A. In the examples of chairs and desks, we observe that "PointBERT-LLaMA" concentrates less on the background (chairs, indicated with red arrows) and more on the actual object surfaces (desks, indicated with red arrows). This demonstrates that the frozen LLM transformer learns to focus on the informative points, which is consistent with our hypothesis.

**2D VQA.**   We investigate the activation of visual tokens in the 2D VQA task in Fig. B. With the METER (Dou et al., 2022) framework initializing the visual backbone from CLIP (Radford et al., 2021) weights, the quality of feature activation is reasonable and it mostly concentrates on the target regions for both the baseline METER and our "METER-LLaMA." However, we can still witness that feature activation better aligns with the images and questions than the noisy attention heads. Furthermore, the activation of "METER-LLaMA" is also slightly advantageous over METER with less scattering, such as better concentrating on the regions of light and leaves at high thresholds.

**3D VQA.**   We analyze the effect of a frozen LLM transformer for 3D VQA and further confirm our hypothesis. As in Fig. C, we compare the activation of visual tokens, which are seed points in

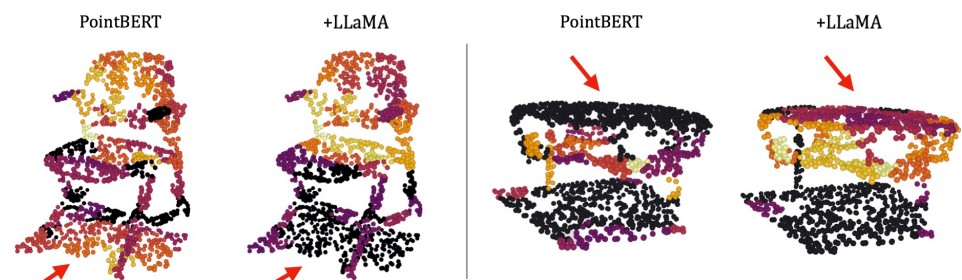

Figure A: Visualization of feature activation for point cloud classification. Brighter colors indicate higher activation values. To highlight the most salient activation, we apply a threshold to filter out points with low activation values. This visualization demonstrates that the model learns to focus on the most discriminative foreground object for classifying the point cloud after adding the frozen LLM transformer. For visual clarity, we use red arrows to indicate the key regions to observe.

| | Attn Scores | Threshold | METER | METER +LLaMA | | Attn Scores | Threshold | METER | METER +LLaMA |
|---|---|---|---|---|---|---|---|---|---|
| Is the light on? | | Low | | | How many leaves? | | Low | | |
| | | High | | | | | High | | |

Figure B: Visualization of attention scores and feature activation for 2D VQA. We are able to visualize attention scores, because METER uses the CLS token. Both low and high thresholds for feature activation are displayed to illustrate the concentration on relevant regions.

VoteNet (Qi et al., 2019), before and after incorporating the LLaMA transformer. To provide the context, the scenes in SQA3D (Ma et al., 2023) are projected onto the bird's-eye-view (BEV). From the visualizations, we clearly observe that the feature activation exhibits sharper concentration on the directions guided by language after adding LLaMA, such as the "table behind me" areas in the left figure and "to my left side" areas in the right figure. Therefore, these indicate that the added LLM transformer selects the informative points and augments them for downstream question answering.

## A.3 QUANTITATIVE EVIDENCE

In Sec. 6.2, we generate the pseudo-masks from feature activation or attention scores and then evaluate their quality with mIoU. This section describes the details.

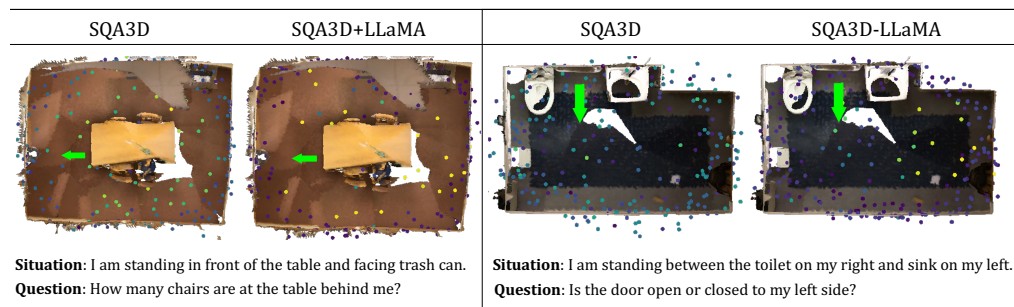

Figure C: Analysis of feature activation for 3D VQA. The scenes are viewed from BEV. The green arrow marks the location and facing direction. The colors of points indicate their activation: a lighter color (yellow) represents a larger magnitude than a dark color (blue and green). We observe that "SQA3D+LLaMA" has sharper activation that is better related to the questions. Thus, it supports our information filtering hypothesis. (**Best viewed in color and zooming in**.)

**Dataset.** We leverage ImageNet-S (Gao et al., 2022) because it provides semantic segmentation masks for ImageNet (Deng et al., 2009) images. Specifically, we adopt the ImageNet-S version with 50 categories and run our evaluation on its validation set, to avoid data leakage from the training set.

**Definition of IoU.** Our IoU calculates the alignment between the highly activated tokens and the ground truth mask. As tokens are sparse and in low resolution, we slightly change the calculation of IoU for our purpose. Specifically, we first project the ground truth mask to the resolution of tokens to acquire a binary mask of tokens, indicating whether they are related to the target object (with value 1) or not (with value 0), denoted as $M_g$. Then we compute the IoU between the pseudo-mask $M_p$ generated from feature activation with $M_g$ as follows:

$$\text{TP} = \text{sum}(M_g M_p), \text{FP} = \text{sum}((1-M_g)M_p), \text{FN} = \text{sum}(M_g(1-M_p)), \quad\quad\text{(E)}$$

$$\widetilde{\text{IoU}}(M_g, M_p) = \text{TP}/(\text{TP}+\text{FP}+\text{FN}). \quad\quad\text{(F)}$$

**Pseudo-mask generation.** To generate pseudo-masks from feature activation, we are motivated by Fig. 3a and treat the highly-activated regions as the final result. To generate pseudo-masks with attention scores, we first sum the scores from all the attention heads and follow a similar procedure of treating highly-scored regions as pseudo-masks. Concretely, given a feature activation $z$, generating a pseudo-mask is as straightforward as $M_p = (z > t)$, where $t$ is a threshold between 0 and 1. Although the process is natural, we notice that selecting the thresholds for activation or score significantly affects the quality of pseudo-masks. Therefore, we always automatically choose the threshold maximizing the mIoU between the pseudo-masks and ground truth masks to avoid threshold tuning and enable a fair comparison. The algorithm of choosing the best threshold for the activation or attention scores for each image is as below,

$$\text{IoU}(M_g, z) = \max_t \left( \widetilde{\text{IoU}}(M_g, M_p) \right), \text{where } M_p = z > t \text{ and } t \in \{0.1, 0.2, 0.3, ..., 0.9\}. \quad\text{(G)}$$

Finally, our mIoU in Sec. 6.2 is the mean IoU on all the images.

**Full results with both magnitude and frequency activation.** Our comparison of mIoU in Fig. 4 only visualizes the larger mIoU from the activation of magnitude or frequency for clarity. To supplement the comprehension, we display the complete statistics in Fig. D. As illustrated, both ViT-S-LLaMA and ViT-B-LLaMA have better pseudo-mask quality from feature activation than attention scores, which directly supports our hypothesis. With the new statistics from both activation types, we additionally notice that the neural network layers have varied preferences over magnitudes or frequencies. However, the ViT-LLaMA features still have better fidelity compared with attention scores and features from ViT. As stated in Sec. 6.2, either magnitude or frequency is an intuitive but lossy way to understand feature representation. Thus, future work is needed to further investigate the advantages and properties of individual layers.

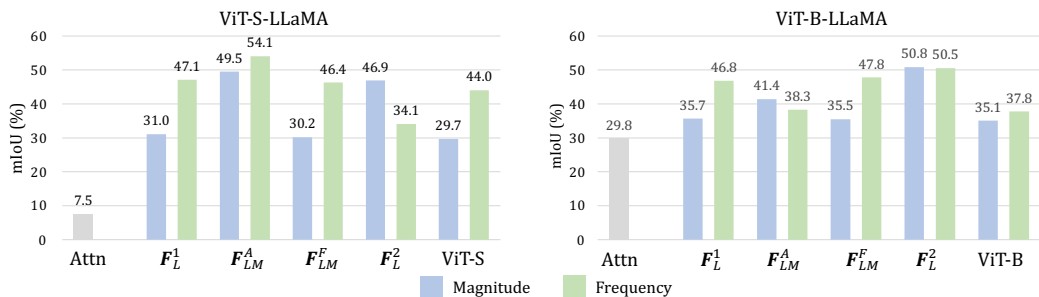

Figure D: Visualization of mIoU between the ground truth masks and pseudo-masks generated from attention scores/feature activation. This figure supplements Fig. 4 by providing the mIoU for both magnitude and frequency activation, where Fig. 4 selects the better one for the clarity of illustration.

### A.4  DISCUSSION AND LIMITATIONS OF THE HYPOTHESIS

**Perspective of usable information.** We supplement the opinions from Xu et al. (2020) that greatly inspire our investigation of the information filtering hypothesis. Xu et al. (2020) propose that a well-trained neural network layer can be considered as a *decipher* adding usable information into the

features and enabling subsequent modules to better infer the latent information. In our information filtering hypothesis, the incorporated modules are indeed acting as *deciphers* that enlarge the contribution of usable tokens and benefit downstream predictions.

**Limitations.** Though our information filtering hypothesis explains how the performance improves with frozen LLM transformers, we notice that several intriguing phenomena are not yet covered. First, the current hypothesis is unable to analyze the utilities of different layers separately. Second, the hypothesis does not explain how the training dynamics facilitate the visual token features to cooperate with the frozen language transformer, which is interesting future work.

### A.5 IMPLEMENTATION DETAILS IN DERIVING THE HYPOTHESIS

**Magnitude and frequency activation**  Our visualization in Sec. 6 and Fig. 3a calculates the feature activation on magnitude or frequency domains to reflect their concentration on target objects. We illustrate the Pytorch-style pseudo-code for our operations in Fig. E. For magnitude, we simply compute the L2 norm of features after centering them. Similarly, the activation in the frequency domain is the norm of the difference between *the angle of a token feature vector* and *the average angle vector of all the tokens within the image* after Fourier transformation. Finally, the activation values are normalized to 0 and 1 for visualization and quantitative analysis.

```python
def activations(visual_tokens):
    # visual_tokens: tensor with shape [H, W, C]
    # magnitude calculation
    avg_token_feature = visual_tokens.mean(dim=0, keepdim=True)
    activation = (visual_tokens - avg_token_feature).norm(dim=-1)
    mag_min, mag_max = activation.min(), activation.max()
    mag_activation = (activation - mag_min) / (mag_max - mag_min)

    # frequency calculation
    freq_token = torch.fft.fft(feat).angle()
    avg_freq_token = freq_token.mean(dim=0, keepdim=True)
    activation = (freq_token - avg_freq_token).norm(dim=-1)
    freq_min, freq_max = activation.min(), activation.max()
    freq_activation = (activation - freq_min) / (freq_max - freq_min)
    return mag_activation, freq_activation
```

Figure E: Pytorch-style pseudo-code for calculating the activation of features on magnitude and frequency domains.

## B  ADDITIONAL ANALYTICAL RESULTS

### B.1  ABLATION STUDY ON DESIGN CHOICES

This section provides supplementary results for the analysis on design choices in Sec. 5.1.

**Results with 100 epochs of training.**  We conduct the ablation studies mostly with 100 epochs to balance the computation and fidelity of conclusions. In addition to the experiments lasting 300 epochs in Sec. 5.1, we supplement them with experiments lasting 100 epochs, summarized in Table A. According to the numbers, adding a frozen LLM transformer as a visual encoder layer is still effective, improving the accuracy of the baseline ViT. In addition, we highlight that fine-tuning is beneficial under insufficient training (100 epochs), but it hurts the accuracy when trained longer due to overfitting, which is analyzed in the next paragraph. In conclusion, both experiments validate our design choices and the effectiveness of using pre-trained LLM transformer blocks as encoder layers.

| Model | Acc |
|---|---|
| *100 Epochs* | |
| ViT-S | 75.3 |
| ViT-S-LLaMA | 75.8 |
| ViT-S-MLP | 75.5 |
| ViT-S-LLaMA-FT | **76.8** |
| *300 Epochs* | |
| ViT-S | 80.1 |
| ViT-S-LLaMA | **80.7** |
| ViT-S-MLP | 80.4 |
| ViT-S-LLaMA-FT | 78.9 |

Table A: Ablation studies on model capacity and fine-tuning.

**Loss curves for fine-tuning.**  We analyze the design choice of fine-tuning the LLM transformer in Sec. 5.1. Fig. F shows the loss curves for both training and validation sets during the training process. The training loss is much larger than the validation loss, because their loss functions are different: the training loss is the label-smoothing cross-entropy, while the validation loss is the regular cross-entropy loss. With the trend in Fig. F, we conclude that jointly fine-tuning the pre-trained LLM transformer might not benefit the performance, yet making

the training process more complicated. Therefore, our experiments adopt the simple solution of freezing the LLM transformer.

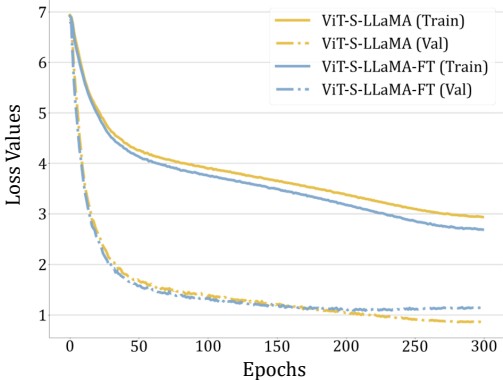

Figure F: Loss curves on the training and validation sets for fine-tuning the LLaMA transformer (ViT-S-LLaMA-FT) or not (ViT-S-LLaMA). We use solid lines to denote training losses and dashed lines to denote validation losses. Though fine-tuning shows advantages in the beginning, it finally hurts the performance due to overfitting (a larger loss on the validation set compared with not fine-tuning).

## B.2 ABLATION STUDY ON NETWORK CAPACITY ACROSS DIVERSE TASKS

This section analyzes whether the improvement of our approach is the consequence of the increased capacity and supplements Sec. 5.1. Specifically, we experiment with two sets of additional baselines: (1) *using additional MLPs*, aligning the number of trainable parameters; (2) *using randomly initialized LLM transformer blocks*, aligning the total number of parameters, to compare with our approach. Please note that the randomly initialized LLM transformer blocks are trained end-to-end with the visual encoders.

We conduct experiments across all the tasks covered in Sec. 4, as shown in Table B. The results show several important findings validating that our performance gains stem from our method rather than the increased network capacity. More importantly, simply adding MLPs is not a consistently beneficial strategy for all the visual tasks and can be detrimental, resulting in inferior performance on some tasks even compared with the plain baselines. This is because naively adding large MLPs or transformer blocks may lead to challenges in optimization and suffers from a small training dataset compared with LLM pre-training.

## B.3 VARYING LLM TRANSFORMER SCALES

This section analyzes the influence of the scales of language transformers with OPT (Zhang et al., 2022). Our experiments incorporate the final transformer layers from OPT-{125M, 350M, 1.3B, 2.7B, 6.7B}, into ViT-S for image classification. Our experiment setting builds upon DeiT (Touvron et al., 2021) and trains for 100 epochs. Additionally, our experiments with small-scale OPT (OPT-{125M,350M}) even yield the loss values of NAN when trained with the original DeiT learning rate, so we decrease their learning rate by 1/5 for stable training. This reflects the importance of the scales of LLMs for stabilizing the training.

| Model | Acc |
|---|---|
| ViT-S | 75.25 |
| + OPT-125M | 71.63 |
| + OPT-350M | 71.56 |
| + OPT-1.3B | 75.62 |
| + OPT-2.7B | 75.74 |
| + OPT-6.7B | 76.29 |

Table C: Accuracy improves with a larger transformer.

As indicated by the results in Table C, the benefits of frozen language transformer grow with increasing capacity of OPT transformers. The added transformers enhance the performance only with sufficient sizes (1.3B, 2.7B, 6.7B) and hurt the accuracy when the sizes are small (125M, 350M). Therefore, the phenomenon of LLM transformers enhancing visual tasks only "emerges" at sufficient scales.

(a) Image Classification (ImageNet)

| Methods | Acc |
|---|---|
| ViT-S | 80.1 |
| + LLaMA (Ours) | **80.7** |
| + MLP | 80.4 |
| + Random LLM | 76.9 |

(b) Point Cloud Recognition (ScanObjectNN)

| Methods | BG | OBJ | T50 |
|---|---|---|---|
| PointBert | 87.4 | 88.1 | 83.1 |
| + LLaMA (Ours) | **88.0** | **88.5** | **83.8** |
| + MLP | 86.5 | 87.3 | 83.4 |
| + Random LLM | 87.2 | 88.0 | 82.6 |

(c) Action Recognition (SSv2)

| Methods | Acc1 | Acc5 |
|---|---|---|
| ViT-S | 64.7 | 89.2 |
| + LLaMA (Ours) | **65.9** | **89.9** |
| + MLP | 63.8 | 88.9 |
| + Random LLM | 64.1 | 88.8 |

(d) Motion Forecasting (Argoverse)

| Methods | ADE↓ | FDE↓ | MR↓ |
|---|---|---|---|
| mmTransformer | 0.72 | 1.10 | 10.7 |
| + LLaMA (Ours) | **0.71** | **1.08** | **10.5** |
| + MLP | 0.74 | 1.16 | 11.8 |
| + Random LLM | 0.74 | 1.15 | 11.5 |

(e) 2D Retrieval (Flickr30k)

| Methods | EM1 | EM5 | EM10 |
|---|---|---|---|
| METER | 49.66 | 80.86 | 89.48 |
| + LLaMA (Ours) | **50.22** | **82.26** | **90.08** |
| + MLP | 49.48 | 81.12 | 89.58 |
| + Random LLM | 49.80 | 81.62 | 89.72 |

(f) 3D VQA (SQA3D)

| Methods | EM1 | EM10 |
|---|---|---|
| SQA3D | 47.20 | 86.82 |
| + LLaMA (Ours) | **48.09** | **89.03** |
| + MLP | 47.14 | 88.08 |
| + Random LLM | 47.26 | 88.46 |

Table B: Addition comparisons for model capacity. We compare our approach of adding the frozen LLM transformer with adding a randomly initialized MLP ("+MLP") or LLM blocks ("+Random LLM") and training end-to-end. The results on diverse tasks uniformly support that our improvement is not merely the result of a larger model capacity. Details are in Sec. B.2.

| Model | Configuration | Acc Top 1 |
|---|---|---|
| ViT-S | Baseline ViT-S | 75.32 |
| + LLaMA at Tail (Ours) | Single LLaMA block at the end of ViT | 75.84 |
| + LLaMA at Middle | Single LLaMA block at the middle of ViT | 75.55 |
| + LLaMA at Head | Single LLaMA block at the head of ViT | 72.66 |
| + 2 LLaMA Blocks | 2 LLaMA blocks at the end of ViT | **77.10** |

Table D: Varying the position and number of LLaMA blocks.

### B.4 LAYERS TO INSERT THE FROZEN LLM BLOCK(S)

Table D presents an ablation study on different architecture variants, including different places to insert the LLM blocks and the number of LLM blocks. Without loss of generality, we leverage the image classification with ViT-S for our ablation study. From the experiments, we mainly have the following discoveries:

**LLM block location.** Inserting the frozen LLaMA block at the beginning or the middle of the visual encoder performs worse than our strategy of inserting LLaMA at the end of the encoder. This supports our design choice and verifies the intuition that LLM blocks are more suitable for high-level semantics instead of low-level visual patterns.

**LLM block number.** We also experiment with inserting the last 2 blocks from LLaMA, and find it to be better than our default strategy of using a single LLaMA block. Due to computation constraints, we are unable to extend this to diverse computer vision tasks as in Sec. 4, but this avenue presents an interesting direction and broadening our understanding of LLMs.

### B.5 BREAKDOWN METRICS FOR 3D VQA

We additionally provide the full metrics on SQA3D in Table E, supplementing Table 5b. As shown in the table, adding a frozen transformer improves the performance on the main metric and most of the analytical metrics.

## C IMPLEMENTATION DETAILS

### C.1 IMAGE CLASSIFICATION

We follow DeiT (Touvron et al., 2021) in training the models of ViT-{T,S,B} and ViT-{T,S,B}-LLaMA in Sec. 4.1. Each visual token is a 16×16 patch on 224×224 images. We adopt the

| Methods | EM@1 | EM@10 | What | Is | How | Can | Which | Others |
|---|---|---|---|---|---|---|---|---|
| ScanQA (Azuma et al., 2022) | 46.58 | 85.97 | 31.64 | 63.80 | 46.02 | 69.53 | 43.87 | 45.34 |
| Multi-CLIP (Delitzas et al., 2023) | 48.02 | - | - | - | - | - | - | - |
| SQA3D (Ma et al., 2023) | 47.20 | 86.82 | 33.48 | 66.10 | 42.37 | **69.53** | 43.02 | **46.40** |
| SQA3D-LLaMA | **48.09** | **89.03** | **34.27** | **67.05** | **48.17** | 68.34 | **43.87** | 45.64 |

Table E: Performance of SQA3D and adding language transformers on 3D question answering (QA). Adding LLaMA achieves the best performance. EM@1 and EM@10 means Top-1 and Top-10 Exact Match (Accuracy) metric. "What," "Is," "How," "Can," "Which," and "Others" are detailed breakdown of question types reported in EM@1.

identical procedure for ViT-T and ViT-S for a fair comparison. The most important configurations include a total of 300 epochs, a base learning rate of 5e-4, a cosine annealing learning rate schedule (Loshchilov & Hutter, 2016), and an AdamW optimizer (Kingma & Ba, 2014; Loshchilov & Hutter, 2017). The total time for training lasts 4-6 days on 4×A100 GPUs. The only change we adopt is the warm-up length of 20 epochs, compared with the original warm-up of 10 epochs in DeiT. A longer warm-up stabilizes the training of ViT models and also enables us to slightly outperform the original ViT-T and ViT-S performance in Table 1.

## C.2    POINT CLOUD CLASSIFICATION

In this section, we describe the implementation details of the point cloud classification method presented in Sec. 4.2. For optimization, we use the AdamW (Kingma & Ba, 2014; Loshchilov & Hutter, 2017) optimizer and a cosine annealing learning rate schedule (Loshchilov & Hutter, 2016). To map the dimension between PointBERT and LLaMa transformer tokens, we add two linear layers with a learning rate of 5e-5. The PointBERT backbone has a learning rate of 5e-4, consistent with the original setting in Yu et al. (2021). We fine-tune our model for 300 epochs on both the ScanObjectNN (Uy et al., 2019) and ModelNet40 (Goyal et al., 2021) datasets. The training takes around 6-10 hours on 4×A100 GPUs

## C.3    ACTION RECOGNITION

We investigate action recognition in Sec. 4.3 and provide more details of implementation here. Our setup strictly follows VideoMAE (Tong et al., 2022), where a ViT model is (1) pre-trained by masked auto-encoding (He et al., 2022); then (2) fine-tuned for additional epochs. As stated in Sec. 4.3, we directly begin from the second step and inherit the parameters of self-attention blocks in ViT from pre-trained VideoMAE models. During the training process, VideoMAE trains ViT-S for 40 epochs (5 epochs of warm-up) and ViT-B for 30 epochs (5 epochs of warm-up), where we adopt the same length of training. The optimizer is AdamW (Kingma & Ba, 2014; Loshchilov & Hutter, 2017) with a cosine annealing learning rate schedule (Loshchilov & Hutter, 2016).

In Sec. 4.3 and Table 3, our ViT-S and ViT-B performance is lower than the reported numbers in VideoMAE. This is because VideoMAE uses 32∼64 GPUs during the fine-tuning stage and supports a much larger batch size compared with ours, though we scale the learning rate according to the batch size as Goyal et al. (2017a). Concretely, VideoMAE adopts the batch size of 384 video clips, while our computational resource only supports a batch size of 24 clips and 12 clips for ViT-S and ViT-B, respectively. However, we control the setup between ViT and ViT-LLaMA for a fair comparison. Finally, the ViT-S-LLaMA and ViT-B-LLaMA experiments take around 3-4 days on 4×A100 GPUs.

## C.4    MOTION FORECASTING

We evaluate the effects of frozen LLM transformers in Sec. 4.4 with motion forecasting. For clarity, we intuitively demonstrate its problem setting and modular architecture in Fig. G, where VectorNet (Gao et al., 2020) or mmTransformer (Liu et al., 2021) encodes each lane or agent trajectory into a token embedding, and then our LLM blocks process these tokens and feed them into the regression decoder.

| Model | ADE↓(k=1) | FDE↓(k=1) |
|---|---|---|
| Paper | 1.66 | 3.67 |
| Ours | 1.60 | 3.60 |

Table F: Our implementation of VectorNet is better than their paper. Please note that this table uses the same single-modal setting (k=1) as VectorNet for a fair comparison.

Since VectorNet and mmTransformer have not released their training code, we reproduce their results on our own and achieve better or similar results as reported in their papers. As in Table F and Table G, the baselines used in our paper (Table 4) have comparable or even better performance compared with their original performance in the papers, which is critical for a fair and meaningful comparison.

During the training time, we separately train VectorNet or mmTransformer. VectorNet is a relatively simple architecture, so its training lasts 60 epochs, with a cosine annealing learning rate schedule (Loshchilov & Hutter, 2016). We use the AdamW optimizer (Kingma & Ba, 2014; Loshchilov & Hutter, 2017) with a learning rate of 5e-4 and a batch size equal to 32 samples. For mmTransformer, we train it with the same learning rate, batch size, and optimizer as VectorNet. The training lasts 32 epochs,

| Model | ADE↓ | FDE↓ | MR↓ |
|---|---|---|---|
| Paper | 0.71 | 1.15 | 10.6 |
| Ours | 0.72 | 1.10 | 10.7 |

Table G: Our implementation of mm-Transformer is comparable to the performance in their paper, with large advantages on the main metric of FDE.

where we drop the learning rate by 1/4 on epochs 20, 24, and 28. The training time for both models is around 2 days on one A100 GPU.

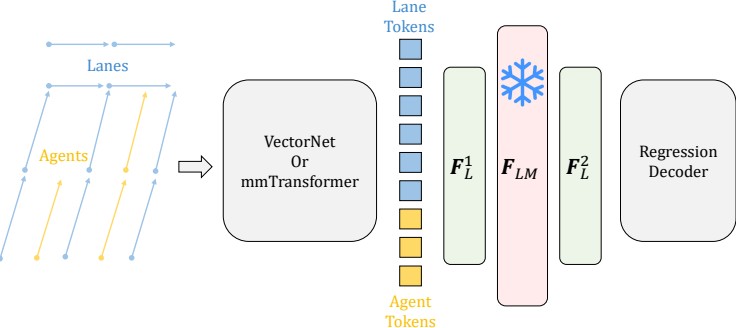

Figure G: Illustration of the typical motion forecasting design and our implementation. Motion forecasting models the trajectories of agents and lanes as polylines. Exiting models (Gao et al., 2020; Liu et al., 2021) use MLPs or transformers to convert the lanes and agent trajectories into token embeddings, and then employ a decoder to regress the future trajectories. In our design, we treat either VectorNet (Gao et al., 2020) or mmTransformer (Liu et al., 2021) as a general encoder, and then insert the frozen LLM blocks to process their embeddings.

## C.5 2D VISION LANGUAGE

This section provides more details on implementation and designs to supplement our discussion on 2D vision-language models in Sec. 4.5. Our experiments adopt the widely-used METER (Dou et al., 2022) as the baseline and incorporate pre-trained LLM transformers after its vision-language fusion module. In Fig. H, we intuitively illustrate the modular design of METER and the specific place to insert our frozen LLM transformer and linear layers.

Conventionally, METER follows a two-stage training strategy: (1) pre-training the whole vision-language model (VLM) on a large combination of vision-language datasets, including COCO (Lin et al., 2014), Conceptual Captions (Sharma et al., 2018), SBU Captions (Ordonez et al., 2011), and Visual Genome (Krishna et al., 2017); (2) fine-tuning on downstream tasks like visual question answering (VQA) or image-text retrieval. However, the first step of pre-training is computationally extensive, so we adopt the setup in Shi et al. (2023) by skipping the pre-training step and directly training on the target task. Specifically, we initialize the image encoder from CLIP-B/32 (Radford et al., 2021) and text encoder from RoBERTa (Liu et al., 2019), and then fine-tune all the modules jointly expect for the LLM transformer $\mathbf{F}_{LM}$. Because of the initialization from CLIP and RoBERTa, our model is capable of predicting reasonably.

During the training stage, we strictly follow the same hyper-parameters and configurations on VQAv2 (Goyal et al., 2017c) and Flickr30k (Plummer et al., 2015) provided by METER. The most critical detail is that METER assigns different learning rates for each module. For example, the cross-modal fusion module and the decoder have larger learning rates compared with the pre-trained

image and text encoders. Similarly, our experiments set the learning rates of linear layers ($\mathbf{F}_L^1$ and $\mathbf{F}_L^2$) $10\times$ the learning rate of the image encoder, because they are randomly initialized. Finally, each training on VQAv2 and Flickr30k lasts for 10 epochs and around 1 day on $4\times$A100 GPUs.

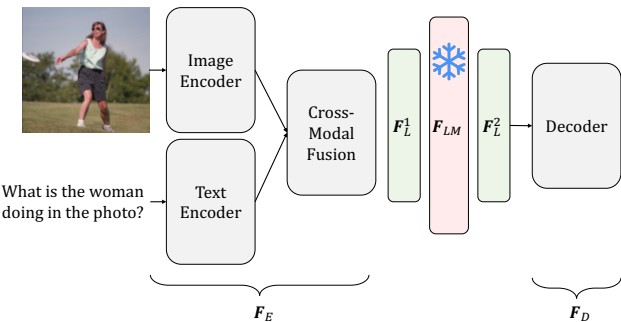

Figure H: Illustration of the METER (Dou et al., 2022) framework and how to insert the frozen language transformer (pink $\mathbf{F}_{LM}$) and linear layers (green $\mathbf{F}_L^1$ and $\mathbf{F}_L^2$) to process the visual tokens after vision-language fusion.

## C.6 3D VISION LANGUAGE

This section provides more details of the dataset and training configurations of the 3D vision-language task. We conduct our experiments on the SQA3D dataset (Ma et al., 2023), which contains $33.4k$ questions in 650 unique ScanNet scenes. In addition to question answering (QA), the benchmark also requires the model to understand its situation (position, orientation, *etc.*) in the 3D scene as described by text. Hence it is called situated question answering (SQA). We use a batch size of 32 during our model training, and AdamW as our optimizer. The hidden size for each embedding token is 768. We train all parameters from scratch for 30 epochs, and decrease the learning rate by 10 times at 10, 15, and 20-th epoch. The model is trained on a single A100 GPU.

## C.7 DEPTHS OF LLM LAYERS

When varying the depth of transformer blocks in Sec. 5.2 and Fig. 2, we adopt the ablation setup of training for 100 epochs, compared with the full training of 300 epochs. The experiments are based on ViT-S/16 (Dosovitskiy et al., 2021) in DeiT (Touvron et al., 2021) with a batch size of 1,024, which is also used in other ablation experiments. The whole training process involves 20 epochs of warm-up and the remaining 80 epochs adopt a cosine annealing learning rate schedule (Loshchilov & Hutter, 2016). Each experiment takes around 2 days on $4\times$A100 GPUs.

