# OpenReview forum: "Frozen Transformers in Language Models Are Effective Visual Encoder Layers"
_ICLR.cc/2024/Conference — ICLR 2024 spotlight_

### Official Review · Reviewer_8b8L · 2023-10-30

**Soundness:** 3 good
**Presentation:** 3 good
**Contribution:** 2 fair
**Rating:** 6
**Confidence:** 4

**Summary:**

This paper suggests a new paradigm for training ViTs and ViT-based multimodal Transformers, which indicates that the performance of ViTs and ViT-based multimodal Transformers can be enhanced by appending a frozen Transformer layer inherited from LLMs behind the original ViTs. Furthermore, this paper develops an information filtering hypothesis to explain how the frozen LLM transformer layer can benefit visual tasks by distinguishing the informative tokens for scenarios directly using all visual tokens and amplifying informative tokens' contributions to [CLS] token for scenarios only using the [CLS] token.

**Strengths:**

- This paper is generally well-written and easy to follow.
- This paper provides detailed investigations of how and why the proposed paradigm works.
- The experimental results verified the proposed paradigm that training ViTs with frozen transformer layers inherited from LLMs is simple yet effective.

**Weaknesses:**

- Method
  - The statement "we place the LLaMA transformer behind the last self-attention block in ViT" is inconsistent with Figure 1. It is unclear why the LLaMA transformer should be placed behind the last self-attention block instead of the last layer (i.e., behind the last FFN block) in ViT. Could the authors provide some insights?

- Experiments
  - Some baselines are missing.
    - For example, it is not persuasive enough to only compare with "ViT-MLP" in Table 6, whose number of trainable parameters remains unchanged, but the total number of the parameters is significantly reduced, especially when layers of LLMs have notably more parameters than layers of ViT -Base/-Small. Therefore, an important baseline that should be added is to append frozen ViT layers behind the original ViT to keep the total number of parameters the same as "ViT-S-LLaMA."
    - Moreover, the total number of parameters, FLOPs, and throughput/latency of ViTs with the LLM layer inserted should be reported to investigate whether the proposed paradigm is more competitive than directly scaling up vanilla ViTs.
  - Some ablation studies are missing.
    - For example, append the frozen LLM layer behind prior layers instead of the last layer in ViT.
    - How does the number of the appended frozen LLM layers affect the model performance?

**Questions:**

See Weaknesses.

---

> ### Author Response · Authors · 2023-11-20
> **Response to Reviewer 8b8L (Part 1/4)**
>
> We appreciate the reviewer for the insightful feedback. We are glad to hear that the reviewer considers our paradigm “simple yet effective” and recognizes our detailed investigation on how it works. Below we address each of the reviewer’s concerns.
>
> ***
>
> 1. *Clarification of our contribution on **diverse** tasks*.
>
> The listed summary and strengths by the reviewer focus on ViT. We would like to kindly clarify that our discovery of “*a frozen LLM transformer transfers to visual tasks*” is not limited to ViT-based architectures, but applicable to a diverse range of tasks and backbones different from ViT, such as point cloud recognition, motion forecasting, and 3D vision-language tasks. Along this line, our major objective is to reveal the surprising generalizability of a frozen LLM transformer, trained only on text data, to a wide range of visual tasks, which is different from aiming to scale up ViT or optimize individual tasks. We hope that this clarification better highlights our contribution and addresses the reviewer’s overall concerns.
>
> ***
>
> 2. *Q: Clarification of method*.
>
> We apologize for the confusion and thank the reviewer for pointing this out. Our method is consistent with Figure 1 (Page 2), i.e., the LLaMA transformer is behind the last FFN block in ViT. We will revise the statement to “behind the last transformer block in ViT” to avoid confusion.

---

> ### Author Response · Authors · 2023-11-20
> **Response to Reviewer 8b8L (Part 2/4)**
>
> 3. *Q: Missing baselines*.
>
> We agree with the reviewer that baselines in addition to ViT-S-MLP will strengthen our paper. However, we are not sure about the suggestion to “append frozen ViT layers behind the original ViT,” because the LLaMA transformers have different parameter numbers compared to ViT transformers. Inferring from the mention of “to keep the total number of parameters the same,” we presume the reviewer is suggesting baselines with equal parameter counts to ViT-S-LLaMA. Therefore, we add the following two sets of experiments and hope they align with what the reviewer suggests. If our interpretation is incorrect, we kindly request the reviewer to provide further clarification, and we are more than willing to address any concerns.
>
> - We first add the experiment of **training from a randomly initialized LLaMA block**, which has the same number of parameters but more trainable parameters. The tables below cover all the tasks investigated in our paper, including image classification, point cloud recognition, action recognition, motion forecasting, 2D vision-language, and 3D vision-language. We will include these results in the revision. The results in the tables show two main findings which validate that the performance gains truly stem from our method: (1) **Our method of adding a frozen LLaMA transformer block consistently outperforms using a randomly initialized transformer block**. (2) **Naively adding a large transformer block can be detrimental for many tasks**, resulting in inferior performance even compared to the plain baseline without the addition of a transformer block. This is because the naive strategy increases the challenges for optimization and suffers from a small training dataset compared to LLM pretraining.
>
> | Image classification   | Acc Top1     | Acc Top5     |
> | ---------------------- | -------- | -------- |
> | ViT-S                  | 80.1     | 95.0     |
> | + LLaMA (Frozen, Ours) | **80.7** | **95.4** |
> | + LLaMA (Random)       | 76.9     | 91.9     |
>
> | Point cloud classification | BG       | OBJ      | T50 (Hardest) |
> | -------------------------- | -------- | -------- | ------------- |
> | PointBERT                  | 87.4     | 88.1     | 83.1          |
> | + LLaMA (Frozen, Ours)     | **88.0** | **88.5** | **83.8**      |
> | + LLaMA (Random)           | 87.2     | 88.0     | 82.6          |
>
> | Action recognition     | Acc Top1     | Acc Top5     |
> | ---------------------- | -------- | -------- |
> | ViT-S                  | 64.7     | 89.2     |
> | + LLaMA (Frozen, Ours) | **65.9** | **89.9** |
> | + LLaMA (Random)       | 64.1     | 88.8     |
>
> | Motion Forecasting     | ADE↓     | FDE↓     | MR↓      |
> | ---------------------- | -------- | -------- | -------- |
> | MMTransformer          | 0.72     | 1.10     | 10.7     |
> | + LLaMA (Frozen, Ours) | **0.71** | **1.08** | **10.5** |
> | + LLaMA (Random)       | 0.74     | 1.15     | 11.5     |
>
> | 2D Vision-language (Zero-shot Retrieval) | EM1       | EM5       | EM10      |
> | ---------------------------------------- | --------- | --------- | --------- |
> | METER                                    | 49.66     | 80.86     | 89.48     |
> | + LLaMA (Frozen, Ours)                   | **50.22** | **82.26** | **90.08** |
> | + LLaMA (Random Init)                    | 49.80     | 81.62     | 89.72     |
>
> | 3D Vision-language (3D VQA) | EM1       | EM10      |
> | --------------------------- | --------- | --------- |
> | SQA3D                       | 47.20     | 86.82     |
> | + LLaMA (Frozen, Ours)      | **48.09** | **89.03** |
> | + LLaMA (Random Init)       | 47.26     | 88.46     |
>
> - The second evidence of aligning with the equal parameter counts is presented in Figure 2 (Page 6) of our original submission. We analyze how the performance of image classification changes with different LLaMA or OPT layers. As shown in the figure, the models can have varied accuracy under the same trainable parameter and total parameter counts. Therefore, it indicates the importance of having proper pretrained weights in the inserted LLM blocks, such as our final LLaMA block, to provide benefits across diverse visual tasks.

---

> ### Author Response · Authors · 2023-11-20
> **Response to Reviewer 8b8L (Part 3/4)**
>
> 4. *Q: Efficiency information*.
>
> First, we agree that sharing detailed configurations is beneficial from an application perspective, and per the reviewer’s request, the table below presents these comparisons. Using a frozen LLM transformer **adds negligible trainable parameters** and **slightly decreases the speed** in image classification, while mainly resulting in an increase in overall parameters and FLOPs.
>
> | Model   | Acc Top1      | Param (Trainable) | Param (All) | FLOPs  | FPS (Batch-Size=1)       | FPS (Batch-Size=256)    |
> | ------- | -------- | ----------------- | ----------- | ------ | ------------------------ | ----------------------- |
> | ViT-S   | 80.1     | 22.05M            | 22.05M      | 8.5G   | 75.26 batches-per-second | 2.54 batches-per-second |
> | + LLaMA | **80.7** | 25.19M            | 227.58M     | 90.11G | 70.18 batches-per-second | 2.36 batches-per-second |
>
> (The FLOPs numbers are evaluated with [`flops_profiler`](https://github.com/cli99/flops-profiler) using a standard image of shape $224\times 224\times 3$. FPS comes from running the inference on 50k ImageNet-val images using batch size of 1 and 256 and a single A6000 GPU).
>
> Second, and more importantly, we would like to clarify the fundamental differences between our work and directly scaling up ViTs. As mentioned earlier, our approach is not tailored for optimizing ViTs, and studies focusing on image classification can develop larger models that surpass our paradigm with proper design and tuning. In contrast, we would like to reiterate that **our goal is to reveal the novel and intriguing discovery** that *despite being trained solely on text data, a frozen LLM transformer can surprisingly function as a visual encoder layer for diverse purely visual tasks, in the absence of language*. Therefore, **our approach is not intended to optimally scale up and enhance individual tasks**, but to reveal that **a frozen set of parameters can be shared across various tasks and backbones**. Even from the perspective of improving models, we would like to clarify:
>
> - Scaling up ViT can lead to improvement in image classification, but **the scaled-up layers may not necessarily extend to diverse tasks as we do**, such as point clouds, videos, and vision-language tasks.
> - To the best of our knowledge, **simply scaling up ViTs does not exhibit the emergent behavior of concentrating on relevant regions as in our information filtering hypothesis**. Furthermore, our approach empirically shows such patterns not limited to images. **Our discovery of this behavior covers videos, point clouds, 2D vision-language, and 3D vision-language tasks**. We summarize three representative results (Figures 3, 5, C from our original submission) in [[this anonymous image link]](https://imgur.com/a/8vEjjqv), for the convenience of checking.

---

> ### Author Response · Authors · 2023-11-20
> **Response to Reviewer 8b8L (Part 4/4)**
>
> 5. *Q: Additional ablation studies*.
>
> We appreciate the reviewer’s excellent suggestions. We would like to again emphasize that our work is **the first** to discover that “*despite being trained solely on text data, a frozen LLM transformer can be used as a visual encoder layer for purely visual tasks, in the absence of language*”. So, **we prioritize validating the general applicability of this discovery across a wide range of visual tasks, in the most straightforward, simplest, and unified manner** – achieved by inserting a single LLM layer at the end of visual backbones. We do not claim that this simple strategy is the optimal approach for achieving the best performance. While we acknowledge that there are various variants to utilize specific LLM layers and insert them into visual backbone architectures which may further improve performance, we deliberately avoid optimizing model selection with more complicated or task-specific architectures.
>
> Following the reviewer’s suggestions, the table below presents an ablation study on different architecture variants. While some variants improve the performance while others decrease the performance, these ablation results contribute to strengthening our understanding and are orthogonal to our main discovery. We will include the results in the revision. And we leave a more comprehensive investigation on these and additional variants as interesting future work.
>
> |      | Model             | Configuration                                               | Acc Top1     |
> | ---- | ----------------- | -------------------------------------------------- | -------- |
> | 1    | ViT-S             | Baseline ViT-S                                     | 75.3     |
> | 2    | + LLaMA           | Single LLaMA block at the end of ViT | 75.8     |
> | 4    | + LLaMA at Middle | Single LLaMA block at the middle of ViT            | 75.5     |
> | 5    | + LLaMA at Head   | Single LLaMA block at the head of ViT              | 72.6     |
> | 6    | + 2 LLaMA Blocks  | 2 LLaMA blocks at the end of ViT                   | **77.1** |
>
> Experimental setting: We conduct the ablation experiments under our ablation setting of the original submission: ViT-S with 100 epochs on ImageNet. This is shorter than a full training of 300 epochs, due to our resource constraints. However, the results in the table are all compared under the identical and fair setup.
>
> - *Q: Inserting more LLM blocks*. Please refer to Row 6 in the table above "+ 2 LLaMA Blocks." We use the final 2 LLaMA blocks, instead of the final single block, to evaluate the impact of using more LLaMA blocks. We find that using more LLM blocks seems beneficial for the overall performance. We think this is an interesting direction to further broaden our discovery and will include it in the revised paper.
>
> - *Q: Other places to insert*. Please refer to Rows 4 and 5 in the table above "+ LLaMA at Middle" and "+ LLaMA at Head." They perform worse than inserting the LLM transformer block at the end of ViT (Row 2), supporting our design choice. Additionally, inserting at the end of encoders is a simple and unified strategy that can be employed across a wide range of tasks and diverse backbones; in contrast, achieving insertion at the middle may require scrutinizing backbones for each task individually.

---

> > ### Comment · Reviewer_8b8L · 2023-11-23
> >
> > I would like to thank the authors for the detailed responses, which have addressed my concerns. I will keep my positive rating.

---

### Official Review · Reviewer_ksvc · 2023-11-01

**Soundness:** 3 good
**Presentation:** 3 good
**Contribution:** 3 good
**Rating:** 6
**Confidence:** 4

**Summary:**

This paper explores the use of a transformer layer from a pre-trained large-language model as part of visual encoders. A frozen transformer block from a pre-trained LLM, typically the last block, is inserted towards the end of standard ViT based visual encoders and this layer is kept frozen throughout the visual encoder training. Experimental results across a range of vision and vision-language benchmarks are presented demonstrating the superiority of the proposed LLM-block enhanced visual encoder against the baseline visual encoder. In explanation to this phenomenon, the paper hypothesizes that LLM-blocks discern important visual information and amplify their effect, and  finally presents qualitative and quantitative results supporting such hypothesis.

**Strengths:**

This paper proposes a simple approach to enhance the capabilities of many existing visual encoders for multiple tasks by just adding frozen LLM blocks with some projection layers into the training process. Experiments are presented on a breadth of image, video, 3D, visual-question answering tasks covering multiple vision problems. Hence, the approach has a broad applicability to the community.

This paper also analyzes why their method works and presents a promising hypothesis: the frozen-LLM selectively amplifies the informative tokens. Fig. 3, 4 and 5 offer evidence to this hypothesis.

The paper is well-written and easy to follow.

**Weaknesses:**

The main weakness of this paper is the choice of the baseline.
- The proposed approach adds an additional (frozen) LLM block and (learnable) projection layers to the baseline network. So, it's unclear whether the performance improvements in Tables 1, 2, 3, 4 and 5 are just because of the increased network capacity.
- The ablation in Table 6 already demonstrates that matching the MLP parameter count already bridges some gap between the proposed method and baseline, revealing a clear inferiority of the baseline used in rest of the tables.
- More importantly, the main (implicit) claim in this paper is that the (frozen) LLM weights have some transferrable knowledge to vision tasks and they can be exploited using the proposed methodology. So, an ideal baseline to support this claim should be the same model with (learnable) weights that are randomly initialized instead of LLM weights.

Without a good baseline, the strong claims across multiple tasks are not well supported.

**Questions:**

What happens to the results in Tables 1, 2, 3, 4 and 5, if we don't use the LLM weights and just use learnable random weights?

---

> ### Author Response · Authors · 2023-11-20
> **Response to Reviewer ksvc**
>
> We appreciate the reviewer for the insightful feedback. We are glad to hear that the reviewer finds our discovery of frozen LLM transformers being visual encoders intriguing, and recognizes our efforts in covering a wide range of tasks and developing the analysis of the information filtering hypothesis. Meanwhile, we understand the reviewer’s concerns about our baselines. Below we address all the points.
>
> Following the reviewer’s suggestions, we conducted additional experiments with two sets of baselines: (1) **using additional MLPs**, aligning the number of trainable parameters; (2) **using randomly initialized LLM transformer blocks**, aligning the total number of parameters, to compare with our approach. The tables below summarize the results for all the tasks investigated in our paper, including image classification, point cloud recognition, action recognition, motion forecasting, 2D vision-language, and 3D vision-language.
>
> We will include these results in the revision. The results in the tables show several important findings which validate that our performance gains stem from our method rather than increased network capacity:
>
> - **Our method of adding a frozen LLaMA transformer block consistently outperforms “-MLP” models**.
> - More importantly, different from the observation in the image classification task (ablation in Table 6 of our submission), **simply adding MLPs is not a consistently beneficial strategy for all visual tasks and can, in fact, be detrimental**, resulting in inferior performance on some tasks even compared to the pain baseline. This is because naively adding large MLPs without the guidance from the pre-trained LLM transformer block may lead to challenges in optimization, especially for modalities and tasks characterized by weaker semantics, such as point clouds and motion forecasting.
> - Similarly, **our method of adding a frozen LLaMA transformer block consistently outperforms using a randomly initialized transformer block**.
> - Similarly, **naively adding a large transformer block can be even detrimental for many tasks**, resulting in inferior performance even compared to the baseline without the addition of a transformer block. This is because the naive strategy increases the challenges for optimization and suffers from a small training dataset compared to LLM pretraining.
>
> | Image Classification  | Acc Top1     | Acc Top5     |
> | --------------------- | -------- | -------- |
> | ViT-S                 | 80.1     | 95.0     |
> | + LLaMA (Ours)        | **80.7** | **95.4** |
> | + MLP                 | 80.4     | 95.0     |
> | + LLaMA (Random Init) | 76.9     | 91.9     |
>
> | Point Cloud Recognition | BG       | OBJ      | T50 (Hardest) |
> | ----------------------- | -------- | -------- | ------------- |
> | PointBERT               | 87.4     | 88.1     | 83.1          |
> | + LLaMA (Ours)          | **88.0** | **88.5** | **83.8**      |
> | + MLP                   | 86.5     | 87.3     | 83.4          |
> | + LLaMA (Random Init)   | 87.2     | 88.0     | 82.6          |
>
> | Action Recognition    | Acc Top1     | Acc Top5     |
> | --------------------- | -------- | -------- |
> | ViT-S                 | 64.7     | 89.2     |
> | + LLaMA (Ours)        | **65.9** | **89.9** |
> | + MLP                 | 63.8     | 88.9     |
> | + LLaMA (Random Init) | 64.1     | 88.8     |
>
> | Motion Forecasting    | ADE↓     | FDE↓     | MR↓      |
> | --------------------- | -------- | -------- | -------- |
> | MMTransformer         | 0.72     | 1.10     | 10.7     |
> | + LLaMA (Ours)        | **0.71** | **1.08** | **10.5** |
> | + MLP                 | 0.74     | 1.16     | 11.8     |
> | + LLaMA (Random Init) | 0.74     | 1.15     | 11.5     |
>
> | 2D Vision-language (Zero-shot Retrieval) | EM1       | EM5       | EM10      |
> | ---------------------------------------- | --------- | --------- | --------- |
> | METER                                    | 49.66     | 80.86     | 89.48     |
> | + LLaMA (Ours)                           | **50.22** | **82.26** | **90.08** |
> | + MLP                                    | 49.48     | 81.12     | 89.58     |
> | + LLaMA (Random Init)                    | 49.80     | 81.62     | 89.72     |
>
> | 3D Vision-language (3D VQA) | EM1       | EM10      |
> | --------------------------- | --------- | --------- |
> | SQA3D                       | 47.20     | 86.82     |
> | + LLaMA (Ours)              | **48.09** | **89.03** |
> | + MLP                       | 47.14     | 88.08     |
> | + LLaMA (Random Init)       | 47.26     | 88.46     |

---

> > ### Comment · Reviewer_ksvc · 2023-11-20
> > **Follow-up question**
> >
> > Thank you for the additional experimentations.
> >
> > Could the authors clarify whether randomly initialized LLAMA block weights are tuned end to end in the above experiments? I’ve suggested such a baseline as a the ideal one.
> >
> > The results in the above tables are very surprising. Especially, when we know that increasing network capacity (i.e. adding additional transformer blocks) helps in some of the above tasks.

---

> > > ### Author Response · Authors · 2023-11-21
> > > **Response to Reviewer ksvc's Follow-up**
> > >
> > > We thank the reviewer for the prompt response and the follow-up question.
> > >
> > > First, yes, the randomly initialized LLaMA block weights **are tuned end to end in the above experiments**.
> > >
> > > Second, we would like to note that the results are not surprising, especially given that in our setting, adding a randomly initialized LLaMA block **significantly increases the trainable parameters**, contrary to what one might expect when adding a ViT transformer layer. This observation also aligns with the findings from prior studies – **naively** adding additional transformer blocks and blindly increasing network capacity does not necessarily improve performance. This is because, as mentioned in our earlier response, “this naive strategy increases the challenges for optimization and suffers from a small training dataset compared to LLM pretraining.”
> > >
> > > To illustrate further, the table below compares the trainable and overall parameter numbers between ViT-S and ViT-S-LLaMA under both our frozen setting and the new baseline of randomly initialized setting.
> > >
> > > |                              | Param (Trainable) | Param (All) |
> > > | ---------------------------- | ----------------- | ----------- |
> > > | ViT-S                        | 22.05M            | 22.05M      |
> > > | + LLaMA (Frozen)             | 25.19M            | 227.58M     |
> > > | + LLaMA (Randomly Initialized) | 227.58M           | 227.58M     |
> > >
> > > From the table, we can see that, for the baseline with a randomly initialized LLaMA block, **the number of trainable parameters increases by more than 10 times**, compared with our frozen model and the plain baseline. Notably, these additional 10 times of parameters all **belong to a single huge LLM transformer block with much larger dimensionality and capacity** than the original ViT backbone. These considerably increased trainable parameters lead to significant optimization difficulties. Moreover, our evaluation consists of a wide range of heterogeneous types of visual tasks with different backbones and different training sets (e.g., lack of large training sets for some tasks). It is expected that such high-capacity models with randomly initialized parameters may not provide consistent benefits and can be even detrimental.
> > >
> > > In a context relevant but different from ours, prior work has shown that **naively adding capacities to a model may not lead to improvement**, because it also needs associated training techniques and datasets. For instance, the authors of SwinTransformer (Liu *et al.*) attempted to scale ViT-B to ViT-L on ImageNet1K in Table 1 of their paper. However, the accuracy dropped from 77.9% to 76.5%. While this observation is not identical to our experimental setting, we provide it here as an additional source of reference that reflects the challenges associated with training large transformers.
> > >
> > > Liu *et al.* Swin Transformer: Hierarchical Vision Transformer using Shifted Windows. ICCV 2021.
> > >
> > > We hope that our explanation has addressed the reviewer’s follow-up question and concern. We are happy to answer any further questions.

---

> > > > ### Comment · Reviewer_ksvc · 2023-11-21
> > > >
> > > > I’d like to thank the authors for providing detailed additional experimentation and answering questions promptly. This has addressed my main concern.
> > > >
> > > > I’ve updated the rating accordingly.

---

> > > > > ### Author Response · Authors · 2023-11-21
> > > > >
> > > > > Thank you for your suggestions and kindly willing to increase the score! We are glad to engage in the insightful discussion with you!

---

### Official Review · Reviewer_sfsm · 2023-11-01

**Soundness:** 3 good
**Presentation:** 3 good
**Contribution:** 3 good
**Rating:** 6
**Confidence:** 4

**Summary:**

The paper proposes to incorporate frozen transformer layers from large language models (LLMs) into vision models when training the model from scratch. By evaluating this approach on a variety of vision tasks, the authors find that when the LLM layer is included, the resulting models observe a slight boost in performance. Finally, the paper proposes the information filtering hypothesis as an explanation to why this happens. Namely, the pre-trained LLM transformer layer focuses the attention and information flow on object-related tokens similar to how self-supervised models such as DINO learn semantically meaningful attention maps.

**Strengths:**

- Reusing pre-trained layers from LLMs is a novel and intriguing idea that could spark further research in the direction of knowledge transfer from foundation models to more specific use cases.
- The experimental evaluation is extensive covering image classification, point cloud detection, video action recognition, motion forecasting, VQA, image retrieval. The proposed method achieves a small boost in performance across the board. The ablation study highlights some important design choices, e.g., which LLM layer to transfer.
- The information filtering hypothesis allows some insight and inspection into the reason for why this method works.
- Good writing style and presentation making the paper an easy read.

**Weaknesses:**

- The experimental improvements are most of the time rather marginal with a few cases showing practical improvements. One could argue that ViT-S-MLP from Tab. 6 would be a fairer comparison for all experiments as it more closely matches the parameter count. In this case, one would expect the gap to further close.
- Some additional ablations might have been of interest.
    - Why only transfer a single LLM layer? Does it also work with more?
    - Why is the layer inserted at the end of the ViT? How does the outcome change when the layer is inserted earlier?
    - How would the results change if we instead transfer a pre-trained vision layer? (see next point)
- The transfer from a language model to a vision model is interesting and most surprising that it works. However, to generalize the approach, one could have also tested to transfer layers from other types of foundation models, e.g. vision-language models (CLIP) or self-supervised vision models (DINO). If there is a considerable gap between LLM and vision layers, one could conclude that, while it works, LLM layers might not be optimal for the domain.
- It remains an open question if this approach scales with large-scale models and training. One could expect diminishing returns as data and model allow for learning equally capable layers. This might still make this approach attractive for smaller scales so it would be interesting to know about this potential limitation. While this is difficult to evaluate, are there any indications how it might behave?

**Questions:**

- Which task and dataset was evaluated for the results in Tab. 6 and Fig. 2?
- The information filtering hypothesis allows some valuable insights. At the same time, it seems to be most prominent after $F_{LM}^A$ judging by Fig. 3, but then in Fig. 4, $F_{LM}^A$ obtains a low score for ViT-B-LLaMA. Why is this the case? Which model size is evaluated in Fig. 3? Is this observation not consistent across architectures?

---

> ### Author Response · Authors · 2023-11-20
> **Response to Reviewer sfsm (Part 1/4)**
>
> We appreciate the reviewer for the thoughtful feedback. We are glad to hear that the reviewer finds our discovery intriguing, covering a wide range of tasks, and the analysis of the information filtering hypothesis insightful. Indeed, we hope that our surprising yet widely applicable findings will motivate new ideas in the community. Below we address each of the reviewer’s concerns.
>
> ***
>
> 1. *Q: Comparing with “-MLP” models*.
>
> We agree that this is a valid concern and conducted additional experiments with the “-MLP” baseline for all the tasks investigated in our paper, including point cloud recognition, action recognition, motion forecasting, 2D vision-language, and 3D vision-language, in addition to the already reported result for image classification, as shown in the tables below. We will include these results in the revision. The results in the tables show two main findings which validate the advantage of our method:
>
> - **Our method of adding a frozen LLaMA transformer block consistently outperforms “-MLP” models**.
> - More importantly, different from the observation in the image classification task, **simply adding MLPs is not a consistently beneficial strategy for all visual tasks and can, in fact, be detrimental**, resulting in inferior performance on some tasks even compared to the pain baseline. This is because naively adding large MLPs without the guidance from the pre-trained LLM transformer block may lead to challenges in optimization, especially for modalities and tasks characterized by weaker semantics, such as point clouds and motion forecasting.
>
> | Image classification (already presented in Table 5 of paper) | Acc Top1     | Acc Top5     |
> | ---------------------------------------------------------- | -------- | -------- |
> | ViT-S                                                      | 80.1     | 95.0     |
> | + LLaMA (Ours)                                             | **80.7** | **95.4** |
> | + MLP                                                      | 80.4     | 95.0     |
>
> | Point cloud recognition | BG       | OBJ      | T50 (Hardest) |
> | ----------------------- | -------- | -------- | ------------- |
> | PointBERT               | 87.4     | 88.1     | 83.1          |
> | + LLaMA (Ours)          | **88.0** | **88.5** | **83.8**      |
> | + MLP                   | 86.5     | 87.3     | 83.4          |
>
> | Action recognition | Acc Top1     | Acc Top5     |
> | ------------------ | -------- | -------- |
> | ViT-S              | 64.7     | 89.2     |
> | + LLaMA (Ours)     | **65.9** | **89.9** |
> | + MLP              | 63.8     | 88.9     |
>
> | Motion forecasting | ADE ↓    | FDE ↓    | MR ↓     |
> | ------------------ | -------- | -------- | -------- |
> | MMTransformer      | 0.72     | 1.10     | 10.7     |
> | + LLaMA (Ours)     | **0.71** | **1.08** | **10.5** |
> | + MLP              | 0.74     | 1.16     | 11.8     |
>
> | 2D Vision-language (Zero-shot Retrieval) | EM1       | EM5       | EM10      |
> | ---------------------------------------- | --------- | --------- | --------- |
> | METER                                    | 49.66     | 80.86     | 89.48     |
> | + LLaMA (Ours)                           | **50.22** | **82.26** | **90.08** |
> | + MLP                                    | 49.48     | 81.12     | 89.58     |
>
> |  3D Vision-language (3D VQA) | EM1       | EM10      |
> | -------------- | --------- | --------- |
> | SQA3D          | 47.20     | 86.82     |
> | + LLaMA (Ours) | **48.09** | **89.03** |
> | + MLP          | 47.14     | 88.08     |

---

> ### Author Response · Authors · 2023-11-20
> **Response to Reviewer sfsm (Part 2/4)**
>
> 2. *Q. Significance of improvements*.
>
> We understand the reviewer’s concerns regarding the significance of our improvements. And we believe that the MLP experiments detailed above have already addressed most of these concerns. Additionally, we would like to emphasize that our objective is not solely focused on improving specific tasks, but rather showing the general applicability of frozen LLM transformers across a wide range of tasks.
>
> Here, we would like to further clarify that **the performance improvements we achieved are significant** for each task, according to the benchmarks established in the published papers within the specific subfield.
>
> - *Image classification*. In our paper (Table 1 on Page 4, Table 6 on Page 6), our ViT-LLaMA model outperforms ViT-S and ViT-S-MLP by 0.6% and 0.3%, respectively. We believe that such improvements are significant for ImageNet1K. To contextualize, we refer to a notable study that includes exhaustive comparison between neural network architectures. ConvNext (Liu *et al.*) compared to its previous state of the art, SwinTransformer (Liu *et al.*) in Table 1, showing a 0.1% improvement for small architectures and a 0.3% for base architectures, which are smaller or comparable to our achieved improvement.
>
> Liu *et al.* A ConvNet for the 2020s. CVPR 2022.
>
> Liu *et al.* Swin Transformer: Hierarchical Vision Transformer using Shifted Windows. ICCV 2021.
>
> - *Point cloud recognition*. In our paper (Table 2 on Page 4), we mainly focus on the challenging ScanObjectNN dataset to evaluate point cloud recognition, where we improve by 0.7% in the hardest T50 setting. As a reference, in the ScanObjectNN's paper (Table 4 in ScanObjectNN, Uy *et al.*), the top-performing methods have the accuracy of 77.9%, 78.1%, and 78.5%, with gaps of 0.2% and 0.4%, which are smaller than our improvement.
>
> Uy *et al.* Revisiting point cloud classification: A new benchmark dataset and classification model on real-world data. ICCV 2019.
>
> - *Action recognition*. We achieve a ~1% improvement on SSv2, which is significant. As a reference, VideoMAE shows a ~0.4% improvement over its previous best method in Table 6 of Tong *et al.*, which is smaller than our gain. This supports that our improvement is significant.
>
> Tong *et al.* VideoMAE: Masked autoencoders are data efficient learners for self-supervised video pre-training. NeurIPS 2022.
>
> - *Motion forecasting.* We improve minADE of mmTransformer from 1.10 to 1.08 (the smaller the better). In mmTransformer's original paper (Table 1, Li *et al.*), its improvement over previous state of the art is also around 0.02. This supports that our improvement is significant.
>
> Liu *et al.*  Multimodal motion prediction with stacked transformers. CVPR 2021.
>
> - *2D Vision-language*. We refer to the METER (Dou *et al.*) paper and VQAv2 as a reference. We improve the performance of METER on VQAv2 by 0.6. According to METER's Tables 2-4, the advantage of selecting the best visual backbone or language encoder also shows a  similar scale of advantage over the remaining models. Hence, we believe that our improvement also significantly benefits the visual encoding.
>
> Dou *et al.* An empirical study of training end-to-end vision-and-language transformers. CVPR 2022.
>
> - *3D Vision-language.* We improve the baseline of SQA3D by around 0.9, while in SQA3D's original paper (Table 3, Ma *et al.*), the best method exhibits a 0.6 advantage over the second one. This verifies the significance of our improvement.
>
> Ma *et al.* SQA3D: Situated question answering in 3d scenes. ICLR 2023.

---

> ### Author Response · Authors · 2023-11-20
> **Response to Reviewer sfsm (Part 3/4)**
>
> 3. *Q: Additional ablation studies*.
>
> We appreciate the reviewer’s excellent suggestions. We would like to first emphasize that our work is **the first** to discover that “*despite being trained solely on text data, a frozen LLM transformer can be used as a visual encoder layer for purely visual tasks, in the absence of language*”. So, **we prioritize validating the general applicability of this discovery across a wide range of visual tasks, in the most straightforward, simplest, and unified manner** – achieved by inserting a single LLM layer at the end of visual backbones. We do not claim that this simple strategy is the optimal approach for achieving the best performance. While we acknowledge that there are various variants to utilize specific LLM layers and insert them into visual backbone architectures which may further improve performance, we deliberately avoid optimizing model selection with more complicated or task-specific architectures.
>
> Following the reviewer’s suggestions, the table below presents an ablation study on different architecture variants. While some variants improve the performance while others decrease the performance, these ablation results contribute to strengthening our understanding and are orthogonal to our main discovery. We will include the results in the revision. And we leave a more comprehensive investigation on these and additional variants as interesting future work.
>
> |      | Model             | Configuration                                               | Acc Top1     |
> | ---- | ----------------- | -------------------------------------------------- | -------- |
> | 1    | ViT-S             | Baseline ViT-S                                     | 75.3     |
> | 2    | + LLaMA           | Single LLaMA block at the end of ViT | 75.8     |
> | 4    | + LLaMA at Middle | Single LLaMA block at the middle of ViT            | 75.5     |
> | 5    | + LLaMA at Head   | Single LLaMA block at the head of ViT              | 72.6     |
> | 6    | + 2 LLaMA Blocks  | 2 LLaMA blocks at the end of ViT                   | **77.1** |
>
> Experimental setting: We conduct the ablation experiments under our ablation setting of the original submission: ViT-S with 100 epochs on ImageNet. This is shorter than a full training of 300 epochs, due to our resource constraints. However, the results in the table are all compared under the identical and fair setup.
>
> - *Q: Inserting more LLM blocks*. Please refer to Row 6 in the table above "+ 2 LLaMA Blocks." We use the final 2 LLaMA blocks, instead of the final single block, to evaluate the impact of using more LLaMA blocks. We find that using more LLM blocks seems beneficial for the overall performance. We think this is an interesting direction to further broaden our discovery and will include it in the revised paper.
>
> - *Q: Other ways to insert*. Please refer to Rows 4 and 5 in the table above "+ LLaMA at Middle" and "+ LLaMA at Head." They perform worse than inserting the LLM transformer block at the end of ViT (Row 2), supporting our design choice. Additionally, inserting at the end of encoders is a simple and unified strategy that can be employed across a wide range of tasks and diverse backbones; in contrast, achieving insertion at the middle may require scrutinizing backbones for each task individually.
>
> ***
>
> 4. *Q: Inserting layers from a visual model*.
>
> Following the reviewer’s suggestion, we use the last transformer layer of CLIP-B/16 and insert it into ViT-S/16 for image classification. The result in the table below shows that this variant does not improve the performance. We hypothesize that this result may indicate that the pre-encoded knowledge in CLIP does not bring in additional useful knowledge for training visual backbones within our current paradigm.
>
> | Model   | Acc Top1      |
> | ------- | -------- |
> | ViT-S   | 75.3     |
> | + LLaMA | **75.8** |
> | + CLIP  | 74.5     |
>
> We appreciate the reviewer’s perspective on interpreting our method as a more general approach to transferring knowledge from any foundation model beyond LLMs. As clarified earlier, our paper focuses on LLM transformers, for the reason exactly pointed out by the reviewer – “the transfer from a language model to a vision model is interesting and most surprising that it works.” Investigating other types of foundation models contributes to strengthening our understanding and is orthogonal to our main discovery on LLM transformers. In addition, the result in the table above suggests that investigating other types of foundation models is not trivial, and we leave this line of exploration as interesting future work.

---

> ### Author Response · Authors · 2023-11-20
> **Response to Reviewer sfsm (Part 4/4)**
>
> 5. *Q: How will it scale*?
>
> We thank the reviewer for the insightful question. Based on our current empirical evaluation, we do not have conclusive evidence indicating the performance gain in scale-up scenarios. Moreover, it seems that this may also vary depending on specific tasks. For example, the improvement from LLM blocks decreases from ViT-Tiny to ViT-Small (Table 1, Page 4) in image classification, but increases from ViT-Small to ViT-Base (Table 3, Page 5) in action recognition. As mentioned by the reviewer, “this is difficult to evaluate,” requiring substantially increased computational resources, and we leave this line of exploration as interesting future work.
>
> On the other hand, we would like to further emphasize that our approach is not intended for an optimal direction to scale up models for **individual tasks**. Instead, it is designed to reveal that **a frozen LLM transformer with unchanged parameters** can surprisingly function as an encoding layer for **diverse visual tasks**.
>
> ***
>
> 6. *Q: Clarification of Tab. 6 and Fig. 2?*
>
> Sorry for the confusion and we will revise the paper to better clarify this. Both Table 6 and Figure 2 (Page 6) are for image classification on ImageNet. The only difference is: Table 6 is trained for the standard 300 epochs, while the ablation in Figure 2 is trained for a more efficient 100 epochs, due to our limited resources. Specifically, fully training one model on ImageNet would occupy most of our GPUs for 3-4 days.
>
> ***
>
> 7. *Q: Questions on the analysis of the information filtering hypothesis?*
>
> The visualization in Figure 3 and the quantitative evaluation in Figure 4 are consistent with our information filtering hypothesis. Below we clarify the specific questions one by one.
>
> - *Q: Which model is evaluated in Fig. 3?* Figure 3 uses ViT-S and ViT-S-LLaMA models for visualization.
>
> - *Q: Why $F_{LM}^A$ has a lower score on ViT-B?* Note that Figure 3 shows the visualization on ViT-S and ViT-S-LLaMA. Now we present additional visualization on ViT-B and ViT-B-LLaMA, shown in [[this anonymous image link]](https://imgur.com/a/XCtD9kz). This image explains why $F_{LM}^A$ in ViT-B-LLaMA has a lower score than ViT-B: **the attention layer $F_{LM}^A$ still clearly distinguishes relevant regions from irrelevant regions**, but the relevant regions are assigned lower activations, which corresponds to a smaller IoU in Figure 4. However, **the illustration in the new ViT-B figure and Figure 4 are still consistent with our information filtering hypothesis**, because the LLM layer finally converts the relevant regions into highly-activated tokens to enlarge their contribution in $F_{LM}^F$ and $F_L^{2}$.
>
> - *Q: Is the observation consistent across architectures?* **Our information filtering hypothesis has consistent observation across architectures**: the features processed by our LLM blocks ( $F_{LM}^F$ and $F_{L}^2$) consistently show a clearer concentration to relevant regions. Meanwhile, we acknowledge that the specific behaviors and utilities of each layer remain an open-problem, *e.g.*, the observed phenomenon that $F_{LM}^A$ can discern tokens in diverse ways.

---

> > ### Comment · Reviewer_sfsm · 2023-11-21
> >
> > I would like to thank the authors for their detailed response answering my questions.
> >
> > The new results are interesting, especially around directly comparing with the MLP variant and the additional ablation on where and how many LLM layers to transfer.
> >
> > > Here, we would like to further clarify that the performance improvements we achieved are significant for each task, according to the benchmarks established in the published papers within the specific subfield.
> >
> > I think some of these comparisons are a bit more nuanced than portrayed. For instance, ConvNext showed that CNNs can achieve competitive classification accuracy on ImageNet despite growing consensus in the community that ViTs are more capable at scale.
> > It is true that for some benchmarks and problems, even a small improvement can be relevant. Significance can also be measured statistically, which, however, requires every paper to publish standard errors and this is not always the case (including for this paper).
> >
> > Overall, the additional information improve the paper, so I strongly encourage the authors to incorporate the promised changes and clarifications into the manuscript.

---

> > > ### Author Response · Authors · 2023-11-22
> > > **Response to Reviewer sfsm's Follow-up**
> > >
> > > Thank you for your response and follow-ups! We are glad to know that our additional experiments have addressed your concerns. As promised in our previous response, we will definitely include all these results and clarifications into our manuscript.
> > >
> > > We understand the reviewer's concern regarding the nuances in evaluating the significance of performance improvements. We acknowledge the complexity of this issue – as pointed out by the reviewer, it is more of a general issue in the community, rather than being specific to our paper. Below, we would like to provide additional clarifications and note our efforts to validate the significance of performance improvements we have achieved. We hope these could address the reviewer’s remaining concern.
> > >
> > > - In our previous response, we **leverage the performance gains reported in representative papers within each subfield as a proxy for the performance variances**, assuming that **each subfield has a consensus on the statistical variance and the reported performance improvement is greater than that the variance**. Take ConvNeXt for instance: ConvNeXt considers its 0.3% improvement over Swin-Transformer as a significant improvement, implying that 0.3% is larger than the variance on ImageNet; thus, our improvement (comparable or exceeding 0.3%) is also greater than the variance and statistically significant. We hope that this line of reasoning resonates with the reviewer – again, a comprehensive evaluation of variance numbers in many computer vision tasks could be challenging and infeasible, often due to the computation restrictions; therefore, the strategy of relying on the consensus of performance gains reported in representative papers is commonly employed in practice.
> > >
> > > - Despite the aforementioned difficulty, we have conducted two experiments on ImageNet (image classification) and Argoverse (motion forecasting). The differences between the two runs are both small for ImageNet in accuracy ($<0.1\%$) and for Argoverse in FDE ($<0.01$). These are smaller than our achieved performance gains. Therefore, we hypothesize that related works do not explicitly report the variance because the scales of variance are commonly recognized as marginal.
> > >
> > > - As a minor note, we would like to point out that ConvNeXt claimed its advantages over previous vision transformers as significant. Quoting from their experiment section, "ConvNeXt also outperforms Swin Transformer of similar complexities across the board."
> > >
> > > We hope that these additional clarifications and explanations resolve the reviewer’s concern on the significance of improvements. We are more than happy to discuss if the reviewer has further questions or comments.

---

> > > > ### Comment · Reviewer_sfsm · 2023-11-23
> > > >
> > > > Thank you for elaborating. I appreciate the effort.
> > > >
> > > > I think being able to show a small improvements across many tasks/datasets as you demonstrate is definitely more convincing as opposed to having marginal improvements on just a few benchmarks.
> > > >
> > > > The rebuttal has addressed my main concerns and questions. I will make a decision about my final score after the reviewer discussion and would like to ask the authors for patience in this regard.

---

### Official Review · Reviewer_SKT3 · 2023-11-02

**Soundness:** 4 excellent
**Presentation:** 3 good
**Contribution:** 3 good
**Rating:** 8
**Confidence:** 4

**Summary:**

Previous work has shown that large language models (LLMs) that are trained only on text can be effectively sutured with visual models via simple linear layers or more complicated cross-attention modules, so as to build multi-modal vision-language frameworks.

This paper studies an interesting problem and tries to answer whether LLMs can deal with single-modal visual tasks without the help of language inputs.

A major finding of this work is that a frozen transformer block from pre-trained LLMs can be used as an effective block in a visual encoder to process visual tokens.

The authors have verified this finding via comprehensive experiences with a variety of 2D and 3D vision tasks.

**Strengths:**

- ***Scope and relevance***: Considering the growing research interest in LLMs, this paper is exceptionally timely as it broadens their application towards visual perception tasks.

- ***Significance of contributions***:  This paper proposed a straightforward yet effective approach to incorporate a frozen transformer block
from a pre-trained LLM as a general-purpose visual encoder layer that can directly process the visual tokens.

- ***Experimental results***: The experiments in this paper are enough to prove the effectiveness of the proposed method and revealed findings.

- ***Clarity***: The main body of the paper is written very well.

**Weaknesses:**

- ***Proposed hypothesis***:  The authors proposed the information filtering hypothesis to explain why pre-trained LLMs can be used for visual encoding. To verify this hypothesis, feature activations regarding both magnitudes and frequencies of features are visualized for comparison. However, visualizing feature activations or attention maps cannot explain their finding well, as transformers (both in ViTs and LLMs) are built with the self-attention mechanism.

- ***Technical details***: Although the inserted LLM block is frozen, is still not clear whether the performance gains truly stem from it. How about adding a random initialized LLM block for comparison?  In Table 2, about the point cloud recognition task, the performance gains are marginal, especially on ModelNet40 (degraded performances for 1k and 4k).

- ***Theoretical justification***: Despite not being compulsory, can the author provide some theoretical justifications rather than experimental observations to explain why the language-trained LLM block can work for visual encoding? Or any potential theory that might lead to a better explanation?

**Questions:**

In addition to the above weaknesses, here are more questions:

- Why only insert one frozen LLM block for visual encoding? can adding more blocks bring more performance again?

- Why choose the last transformer block from LLaMA-7B for experiments? Can other blocks also work well? Is there any guidance to choose which block?

- Is there any other way to insert the frozen LLM block in visual encoders? How about adding it at the beginning of visual transformers, or in the middle of visual transformers?

- The authors removed the attention mask and positional embedding for the frozen LLM block, is there any experimental study for these designs?

---

> ### Author Response · Authors · 2023-11-20
> **Response to Reviewer SKT3 (Part 1/4)**
>
> We appreciate the reviewer for the thorough feedback and the recognition of our contribution,  which generalizes pretrained LLM transformers to diverse visual tasks and introduces the information filtering hypothesis as explanation. Below we address each of the reviewer’s concerns.
>
> ***
>
> 1. *Q: Feature visualization clarification*.
>
> We would like to first clarify that visualizing feature activations and attention maps is a widely-adopted strategy in related studies to investigate the emergent behaviors of ViT features, such as in DINO (Caron *et al.*) and AbsViT (Shi *et al*.). This is because these visualizations represent the most direct output and intermediate variables from transformers. In this work, we followed a similar strategy to demonstrate how features concentrate on relevant regions with our frozen LLM transformer, providing empirical validation for our information filtering hypothesis. Importantly, our visualizations extend beyond those conducted in DINO and AbsViT, by encompassing diverse tasks (videos, 3DVQA, etc.) and introducing the “frequency” activation to better capture the patterns of features. If the reviewer has alternative suggestions for visualization and analysis methods, we are happy to discuss and utilize them.
>
> Caron *et al.* Emerging Properties in Self-Supervised Vision Transformers. ICCV 2021.
>
> Shi *et al*. Top-Down Visual Attention from Analysis by Synthesis. CVPR 2023.
>
> ***
>
> 2. *Q: Training from a randomly initialized LLM block*.
>
> In the tables below, we present the ablation studies on *training from a randomly initialized LLM block* for all the tasks investigated in our paper, including image classification, point cloud recognition, action recognition, motion forecasting, 2D vision-language, and 3D vision-language. We will include these results in the revision. The results in the tables show two main findings which validate that the performance gains truly stem from our method:
>
> - **Our method of adding a frozen LLaMA transformer block consistently outperforms using a randomly initialized transformer block**.
> - **Naively adding a large transformer block can be detrimental for many tasks**, resulting in inferior performance even compared to the baseline without the addition of a transformer block. This is because the naive strategy increases the challenges for optimization and suffers from a small training dataset compared to LLM pretraining.
>
> | Image classification   | Acc Top1     | Acc Top5     |
> | ---------------------- | -------- | -------- |
> | ViT-S                  | 80.1     | 95.0     |
> | + LLaMA (Frozen, Ours) | **80.7** | **95.4** |
> | + LLaMA (Random Init)  | 76.9     | 91.9     |
>
>
> | Point cloud classification | BG       | OBJ      | T50 (Hardest) |
> | -------------------------- | -------- | -------- | ------------- |
> | PointBERT                  | 87.4     | 88.1     | 83.1          |
> | + LLaMA (Frozen, Ours)     | **88.0** | **88.5** | **83.8**      |
> | + LLaMA (Random Init)      | 87.2     | 88.0     | 82.6          |
>
> | Action recognition     | Acc Top1     | Acc Top5     |
> | ---------------------- | -------- | -------- |
> | ViT-S                  | 64.7     | 89.2     |
> | + LLaMA (Frozen, Ours) | **65.9** | **89.9** |
> | + LLaMA (Random Init)  | 64.1     | 88.8     |
>
> | Motion Forecasting    | ADE↓     | FDE↓     | MR↓      |
> | --------------------- | -------- | -------- | -------- |
> | MMTransformer         | 0.72     | 1.10     | 10.7     |
> | + LLaMA (Frozen, Ours)  | **0.71** | **1.08** | **10.5** |
> | + LLaMA (Random Init) | 0.74     | 1.15     | 11.5     |
>
> | 2D Vision-language (Zero-shot Retrieval) | EM1       | EM5       | EM10      |
> | ---------------------------------------- | --------- | --------- | --------- |
> | METER                                    | 49.66     | 80.86     | 89.48     |
> | + LLaMA (Frozen, Ours)  | **50.22** | **82.26** | **90.08** |
> | + LLaMA (Random Init)                    | 49.80     | 81.62     | 89.72     |
>
> | 3D Vision-language (3D VQA) | EM1       | EM10      |
> | --------------------------- | --------- | --------- |
> | SQA3D                       | 47.20     | 86.82     |
> | + LLaMA (Frozen, Ours) | **48.09** | **89.03** |
> | + LLaMA (Random Init)       | 47.26     | 88.46     |

---

> ### Author Response · Authors · 2023-11-20
> **Response to Reviewer SKT3 (Part 2/4)**
>
> 3. *Q: Point cloud recognition performance on ModelNet40*.
>
> We discussed and explained the marginal performance gain on ModelNet40 in the last paragraph of Page 4 in the original submission. ModelNet40 is an earlier dataset in point cloud classification, and the performance on it has saturated in recent years (Ma *et al.*). Hence, the observed slight performance drop (~0.2%) in the easier settings (1k, 4k points) is within the norm. Given this context, we concentrated on the more challenging and recent ScanObjectNN dataset, Table 2 (Page 4), for a more informative comparison. On ScanObjectNN, our approach achieves consistent improvements as shown in the table below (copied from Table 2 on Page 4), especially in the hardest T50 setting.
> |                        | BG              | OBJ (Easiest)   | T50 (Hardest)   |
> | ---------------------- | --------------- | --------------- | --------------- |
> | PointBERT              | 87.4            | 88.1            | 83.1            |
> | + LLaMA (Frozen, Ours) | **88.0** (+0.6) | **88.5** (+0.4) | **83.8** (+0.7) |
>
> Ma *et al.* Rethinking network design and local geometry in point cloud: A simple residual MLP framework. ICLR 2022.
>
> ***
>
> 4. *Q: Theoretical justification*.
>
> In Sec. A.4 (Page 16) of the original submission, we discussed a potential theory that supports our information filtering hypothesis – **“usable information theory”** (Xu *et al.*). Specifically, Xu *et al.* posit that a well-trained neural network can be treated as a *decipher* and increases the usable information in latent features, so that subsequent layers can translate features into correct predictions/labels more easily. We think this theory aligns well with our observation: a pretrained LLM block amplifies the relevant tokens and naturally enables the decoder to predict better labels.
>
> Additionally, we would like to point out that understanding LLMs and the training mechanism of neural networks remains open problems; so our paper primarily focuses on empirical discoveries and investigations. We leave a more in-depth theoretical analysis as interesting future work. Meanwhile, we hope that our information filtering hypothesis can offer additional perspectives to understand LLM transformers.
>
> Xu *et al.* A theory of usable information under computational constraints. ICLR 2020.

---

> ### Author Response · Authors · 2023-11-20
> **Response to Reviewer SKT3 (Part 3/4)**
>
> 5. *Q: Additional ablation studies*.
>
> We appreciate the reviewer’s excellent suggestions. We would like to first emphasize that our work is **the first** to discover that “*despite being trained solely on text data, a frozen LLM transformer can be used as a visual encoder layer for purely visual tasks, in the absence of language*”. So, **we prioritize validating the general applicability of this discovery across a wide range of visual tasks, in the most straightforward, simplest, and unified manner** – achieved by inserting a single LLM layer at the end of visual backbones. We do not claim that this simple strategy is the optimal approach for achieving the best performance. While we acknowledge that there are various variants to utilize specific LLM layers and insert them into visual backbone architectures which may further improve performance, we deliberately avoid optimizing model selection with more complicated or task-specific architectures.
>
> Following the reviewer’s suggestions, the table below presents an ablation study on different architecture variants, together with some ablation provided in the original submission. While some variants improve the performance while others decrease the performance, these ablation results contribute to strengthening our understanding and are orthogonal to our main discovery. We will include the results in the revision. And we leave a more comprehensive investigation on these and additional variants as interesting future work.
>
> |     | Model             | Configuration                                               | Acc Top1     |
> | ---- | ----------------- | -------------------------------------------------- | -------- |
> | 1    | ViT-S             | Baseline ViT-S                                     | 75.3     |
> | 2    | + LLaMA           | Single LLaMA block at the end of ViT | 75.8     |
> | 4    | + LLaMA at Middle | Single LLaMA block at the middle of ViT            | 75.5     |
> | 5    | + LLaMA at Head   | Single LLaMA block at the head of ViT              | 72.6     |
> | 6    | + 2 LLaMA Blocks  | 2 LLaMA blocks at the end of ViT                   | **77.1** |
>
> Experimental setting: We conduct the ablation experiments under our ablation setting of the original submission: ViT-S with 100 epochs on ImageNet. This is shorter than a full training of 300 epochs, due to our resource constraints. However, the results in the table are all compared under the identical and fair setup.
>
> - *Q: Inserting more LLM blocks*. Please refer to Row 6 in the table above "+ 2 LLaMA Blocks." We use the final 2 LLaMA blocks, instead of the final single block, to evaluate the impact of using more LLaMA blocks. We find that using more LLM blocks seems beneficial for the overall performance. We think this is an interesting direction to further broaden our discovery and will include it in the revised paper.
>
> - *Q: Other ways to insert*. Please refer to Rows 4 and 5 in the table above "+ LLaMA at Middle" and "+ LLaMA at Head." They perform worse than inserting the LLM transformer block at the end of ViT (Row 2), supporting our design choice. Additionally, inserting at the end of encoders is a simple and unified strategy that can be employed across a wide range of tasks and diverse backbones; in contrast, achieving insertion at the middle may require scrutinizing backbones for each task individually.
>
> - *Q: Other LLM blocks*. In the original submission, we analyzed different blocks from LLaMA and OPT in Fig. 2 (Page 6), under the same ablation setting. We found that different transformer blocks from LLaMA or OPT can improve image classification accuracy. Note that it is time and resource-prohibitive for us to try all possible combinations across all of our tasks. Hence, we simply chose the last block from LLaMA-7B, because it is sufficient for us to convey our discovery. As for the guidance of selecting blocks, our observation from Fig. 2 indicates that the final few blocks from LLM consistently lead to improvement.

---

> ### Author Response · Authors · 2023-11-20
> **Response to Reviewer SKT3 (Part 4/4)**
>
> 6. *Q: Attention mask and positional embedding*.
>
> We would like to clarify that **our approach uses the attention mask strategies and positional embeddings identical to those used in the corresponding visual tasks**. We do not use the auto-regressive attention masks and rotary positional embeddings designed for LLMs, since they are not directly applicable to visual tasks, and thus we do not include experiments for this. Specifically, most visual models do not use auto-regressive attention masks or rotary positional embedding, because their tokens are *2- or 3-dimensional* and should be *considered all at once*, compared to a *1-dimensional temporal sequence* in language. For example, ViT (Dosovitski *et al.*) for classification uses learnable positional embedding and no attention masks; and VectorNet (Gao *et al.*) for motion forecasting encodes 3D coordinates into features and only uses attention masks to indicate padded tokens. Therefore, we directly follow the common practice of each visual task.
>
> Dosovitski *et al.* An image is worth 16x16 words: Transformers for image recognition at scale. ICLR, 2021.
>
> Gao *et al.* Vectornet: Encoding hd maps and agent dynamics from vectorized representation. CVPR, 2020.

---

> > ### Comment · Reviewer_SKT3 · 2023-11-23
> >
> > I would like to thank the authors for their detailed clarification, which has addressed my concerns about their work. Therefore, I raise my rating to accept.

---

### Author Response · Authors · 2023-11-20
**General Response to All**

We are thankful for the feedback and suggestions from all the reviewers. We are glad that the reviewers recognize our intriguing, exceptionally timely discovery that a frozen LLM transformer can transfer to visual encoding (SKT3, sfsm, ksvc, 8b8L), the associated novel, straightforward yet effective approach (SKT3, sfsm, 8b8L), and the wide breadth of visual tasks investigated in the paper (SKT3, sfsm, ksvc) supported by extensive experiments (SKT3, sfsm, 8b8L). Additionally, our proposed *information filtering hypothesis* is insightful and promising (sfsm, ksvc), which can spark further research (sfsm).

We address each of the reviewers’ concerns in the individual response. Here, we would like to highlight the key objectives and contributions of our paper:

- Being the first work, we discover and share the intriguing phenomenon: **the transformers from LLMs, despite being trained solely on text data, can surprisingly function as encoder layers for purely visual tasks, in the absence of language**. Instead of optimizing this paradigm for a single or few tasks, we primarily promote the breadth and generalizability of our discovery. We achieve this by employing the most straightforward, simplest, and unified design across a wide range of tasks, including image classification, point cloud classification, action recognition from videos, motion forecasting, 2D VQA and image-text retrieval, and 3D VQA from point clouds.

- We discover the **information filtering hypothesis** emergently to explain the observed improvement from a frozen LLM transformer: the LLM transformer **distinguishes the informative tokens** and **amplifies their contribution**, in the form of enlarged magnitudes or frequencies in the feature activation. Such emergent behaviors of separating relevant features were rarely observed before in supervised visual models. More surprisingly, our approach empirically shows this behavior across a wide range of tasks and modalities, spanning images, point clouds, videos, and 2D/3D vision-language tasks. (We summarize three representative results from Figure 3, 5, C of our original submission in [[this anonymous image link]](https://imgur.com/a/8vEjjqv), for the convenience of checking.) We highlight that such investigation and analysis is one of the major contributions of our paper and will likely spark new insights in the community.

Therefore, we kindly suggest that our contributions aim at presenting the applicability of our discovery and the intriguing emergent behaviors across a wide range of tasks, instead of focusing on scaling up and optimizing individual tasks. We are grateful that reviewers (SKT3, sfsm) approach our paper from this perspective.

In addressing the reviewers’ main concerns, we provide additional experiments in the individual responses, validating that 1) our performance gains originate from our method rather than increased network capacity; 2) we consistently achieve performance improvements across tasks, compared to the inconsistent and even detrimental behaviors observed with alternative methods of *using additional MLPs* and *using randomly initialized LLM transformer blocks*.

---

### Meta-Review · Area_Chair_Y9Q8 · 2023-12-12

**Metareview:**

The paper proposes to incorporate frozen transformer layers from large language models (LLMs) into vision models when training the model from scratch. After rebuttal, this paper received scores of 6668.

All the reviewers are happy about the paper, commenting that the paper is well written, it provides detailed investigations of how and why the proposed paradigm works, reusing pre-trained layers from LLMs is a novel and intriguing idea, and experimental results are sufficient. The information filtering hypothesis also provides some insights into the reason for why this method works. Therefore, the AC would like to recommend acceptance of the paper.

**Justification For Why Not Higher Score:**

The improvements of the proposed approach are rather marginal making this paper conceptually interesting, but less appealing practically, thus not higher score.

**Justification For Why Not Lower Score:**

All the reviewers are positive about the paper, and found the findings in the paper intriguing. Experiments are comprehensive to show the effectiveness of the method.

---

### Decision · Program_Chairs · 2024-01-16

Accept (spotlight)